

# Large-scale flood risk assessment in data scarce areas: an application to Central Asia

Gabriele Coccia[1], Paola Ceresa[1], Gianbattista Bussi[1], Simona Denaro[1], Paolo Bazzurro[1,2],
Mario Martina[1,2], Ettore Fagà[1], Carlos Avelar[3], Mario Ordaz[3], Benjamin Huerta[3], Osvaldo
Garay[3], Zhanar Raimbekova[4], Kanatbek Abdrakhmatov[5], Sitora Mirzokhonova[6], Vakhitkhan
Ismailov[7], Vladimir Belikov[8]

[1]Risk Engineering + Development (RED), Pavia, 27100, Italy
[2]Scuola Universitaria Superiore Pavia (IUSS), Pavia, 27100, Italy
[3]Evaluación de Riesgos Naturales (ERN), Mexico City, 01050, Mexico
[4]Department of Geography and Environmental Sciences, Al-Farabi Kazakh National University, A15E3C7, Al-Farabi ave., 71/19, Almaty, Kazakhstan
[5]Institute of Seismology of the National Academy of the Kyrgyz Republic, Bishkek, 720060, Kyrgyz Republic
[6]Institute of Water Problems, Hydropower Engineering and Ecology, Dushanbe, 734042, Tajikistan
[7]Institute of the Seismology of the Academy of Sciences of Uzbekistan, Tashkent, 700128, Uzbekistan
[8]Independent consultant, Ashgabat, 744000, Turkmenistan
*Correspondence to*: Paola Ceresa (paola.ceresa@redrisk.com)



**Abstract**

The countries of Kazakhstan, Kyrgyz Republic, Tajikistan, Turkmenistan and Uzbekistan in Central Asia are highly prone to natural hazards, more specifically, floods, earthquakes, and landslides. The European Union, in collaboration with the World Bank and the GFDRR, created the program "Strengthening Financial Resilience and Accelerating Risk Reduction in Central Asia" (SFRARR), aiming to advance disaster and climate resilience in Central Asia. Within the framework of the SFRARR project, the "Regionally consistent risk assessment for earthquakes and floods and selective landslide scenario analysis for strengthening financial resilience and accelerating risk reduction in Central Asia" was conceived to help handle and achieve the parent project objectives.

A fully probabilistic risk assessment for pluvial and fluvial floods has been carried out for Kazakhstan, Kyrgyz Republic, Tajikistan, Turkmenistan and Uzbekistan for supporting regional and national risk financing and insurance applications, including potential indemnity and/or parametric risk financing solutions for the structuring of a regional program. The pluvial flood part of the study, however, is omitted here for brevity. A homogenized risk assessment methodology for the five countries and across multiple hazards (floods and earthquake) and asset types has been adopted to obtain strategic financial solutions consistent across geographical areas and across economic sectors.

This article presents the data, model, methodology and results for the five Central Asia countries of the flood risk assessment, which represents the first high-resolution regional-scale transboundary risk assessment study in the area aiming at providing tools for decision-making. The output information will inform and enable the World Bank to initiate a policy dialogue.

Currently, the availability of risk information for Disaster Risk Management (DRM) and Disaster Risk Financing and Insurance (DRFI) activities remains variable across the region and has been provided by previous projects focusing on a single country. Moreover, few of these studies have quantified multi-hazard disaster risk, and, to our best knowledge, none have done so for the whole region using probabilistic methods applied with the sufficient fidelity required to robustly inform the development of DRFI solutions.

*Keywords*: flood hazard, flood risk, large-scale model, risk assessment, Central Asia

**Short summary**

A probabilistic flood risk assessment was carried out for five Central Asia countries for supporting regional and national risk financing and insurance applications. A homogenized risk assessment methodology for the countries and across multiple hazards (floods and earthquake) and asset types was adopted to obtain strategic financial solutions consistent across geographical areas and across economic sectors. The paper presents the first high-resolution regional-scale transboundary flood risk assessment study in the area aiming at providing tools for decision-making.

# 1 Introduction

Central Asia is subject to frequent disasters including earthquakes and floods (GFDRR, 2016). Furthermore, climate change, urbanization processes, and a growing population have contributed to an increase in the frequency and severity of losses caused by natural hazard events in the last two decades (Pollner et al., 2010; Yu et al., 2019; Reyer et al., 2017). The transboundary nature of many of these events requires a regional and shared approach to support, plan, and coordinate Disaster Risk Management (DRM) and Disaster Risk Financing and Insurance (DRFI) strategies. Flood risk assessment is a fundamental tool in this framework, as it allows the quantification of the expected losses caused by floods and the identification and prioritization of interventions (Tsakiris, 2014; Merz et al., 2014). Flood risk assessment is defined as an evaluation of future losses caused by floods (riverine and/or coastal) using a set of tools such as hydrological and flood models, exposure models and vulnerability models within a risk-based framework, which includes associating losses with levels of likelihood (Mitchell-Wallace et al., 2017).

In particular, large-scale risk assessment is needed by governments and international institutions to drive national-scale policies to counter economic losses by floods and improve national resilience towards disasters caused by natural hazard events. Here, we define large-scale risk assessment as a risk evaluation study that covers an area


encompassing hundreds of thousands of square km, including administrative units from districts/provinces to national or pluri-national scale. Large-scale flood hazard modelling and assessment is nowadays a well-established branch of flood engineering research and practice (Alfieri et al., 2014; Pappenberger et al., 2012; Schumann et al., 2016), albeit with caveats and limitations (Bates, 2022). Large-scale flood risk modelling and

assessment has also gained traction in the past years (Steinschneider et al., 2014; Ward et al., 2013) and is routinely used in commercial catastrophe risk models by insurance and reinsurance companies to price their products (Wing et al., 2020). Nevertheless, uncertainties remain large and their evaluation is subject of ongoing research (Figueiredo et al., 2018).

A key issue for large-scale model set-up and reliability is data availability. Such models are data-demanding, since

they need meteorological data, river flow observations, geomorphological data, location and protection level of defences, macroeconomic data, among others. Such data might not always be available, or cannot be easily obtained, due to data restriction policies or lack of digitalisation. In Central Asia, for example, meteorological and flow data are hard to acquire without an institutional or local support. Furthermore, in this region, several flow gauges were discontinued at the time of the dissolution of the Soviet Union, and most of them were not replaced

and, therefore, flow records covering recent times are scarce. Another frequent limitation is the absence of post-event survey, either of the event intensity (flood footprints) or of the physical damage and of the economic losses (e.g., damage data and insurance claims).

In this study, a flood risk assessment model was implemented based on regional and local datasets, which comprise a hazard module (assessment of frequency and intensity of floods), a vulnerability module (assessment of the

relationship between event intensity and damage/losses) and an exposure module (inventory of building and infrastructure). The model covers the countries of Kazakhstan, Kyrgyz Republic, Tajikistan, Turkmenistan and Uzbekistan, in Central Asia. The model was implemented within the framework of the project "Regionally Consistent Risk Assessment for Earthquakes and Floods and Selective Landslide Scenario Analysis for Strengthening Financial Resilience and Accelerating Risk Reduction in Central Asia" and within the

implementation of the EU-Funded "Strengthening Financial Resilience and Accelerating Risk Reduction in Central Asia" – SFRARR program (https://www.gfdrr.org/en/program/SFRARR-Central-Asia). The project aims to advance disaster and climate resilience in Central Asia countries. The landslide susceptibility assessment, which was part of this study, can be found in (Rosi et al., 2023).

The objective of this paper is twofold. First to provide guidelines for the implementation of large-scale (e.g.,

country-scale) flood risk models in data-scarce regions, showing that regional datasets such as reanalysis and global land maps need to be integrated with knowledge and data that can only be obtained through engagement and collaboration of local authorities and local experts through participation of stakeholders. Second to present for the five countries considered in this study the estimated levels of flood risk to support governments and decision makers in the decisions for a more comprehensive flood risk management strategy. Fragmented and low-

resolution flood risk assessment studies already exist in the region (CAC DRMI, 2009; GFDRR, 2016; UNDP, 2014; UNISDR, 2010; Umaraliev et al., 2020; Asian Development Bank, 2015; M.S. Saidov, 2020), however a high-resolution, homogeneous, transboundary flood risk assessment such as the one presented here is unprecedented for the Central Asia region.

## 2 Study area

Central Asia is highly exposed and vulnerable to a broad range of natural hazards which frequently result in economic and human losses. Flood hazard is significant in the region, with floods being the most frequent natural disaster in the period 1988-2007 according to a recent analysis provided by the Central Asia and Caucasus Disaster Risk Management Initiative (CAC DRMI, 2009). In the same period, floods were second for number of deaths caused and amount of population affected (1,512 and 19% respectively). Despite the aridity of large areas in some

of the target countries, natural phenomena linked to extreme precipitation can cause billions of dollars of damages every year: collectively, floods inflict the second highest overall economic losses ($52 million), surpassed only by earthquake (an annual average of $186 million). At the local level (e.g., in Tajikistan), flood is sometimes the dominant risk in terms of economic losses (World Bank et al., 2012). Considering the deteriorated protection infrastructure and vulnerabilities in several sectors, floods can cause considerable damage to housing,

infrastructure, and agriculture (Libert, 2008).

Climatically, this region is characterized by strong rainfall gradient contrasts, due to the diversity of climate and vegetation zones. The region is drained by large, partly snow- and glacier-fed mountain rivers, that cross or terminate in arid forelands. Central Asian countries are therefore affected by a significant river flood hazard mainly in spring and summer seasons. Land use is mainly grassland in central and southern Kazakhstan, while in most of Uzbekistan and Turkmenistan vegetation is very sparse. Arable land is concentrated in northern Kazakhstan and in the irrigated parts of the plains of Uzbekistan and Turkmenistan. Tajikistan and the Kyrgyz Republic are mainly mountainous while the other three countries are mostly flat. The elevation of the region is shown in Figure 1.

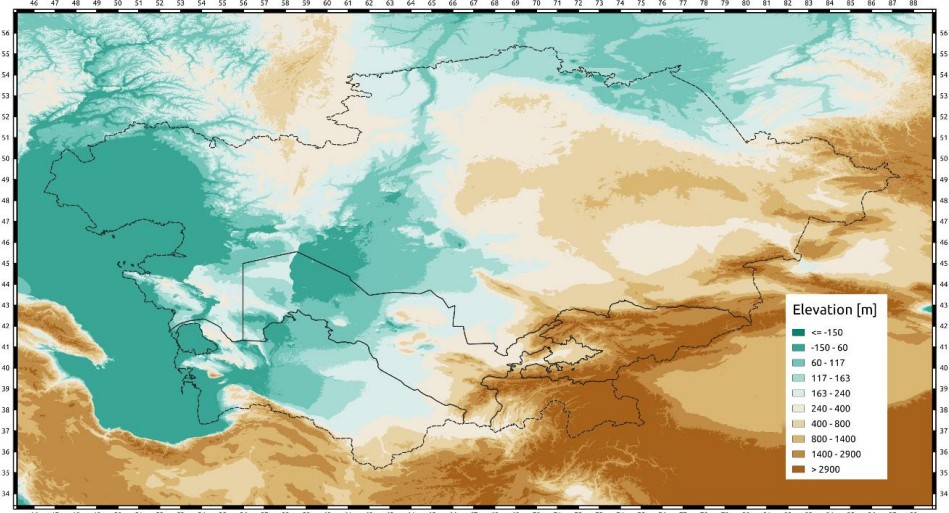

**Figure 1. Study area: country boundaries and elevation.**

## 3 Data and models

### 3.1 Global datasets

We utilized the ECMWF ERA5-Land (Muñoz-Sabater et al., 2021; Hersbach et al., 2019) and the KNMI Climate Explorer (Trouet and Van Oldenborgh, 2013) datasets. ERA5-Land provides hourly 0.1°×0.1° estimates for both precipitation and temperature and combines model data with observations from across the world into a globally complete and consistent dataset. ERA5 is updated daily with a latency of about 5 days. At the time of the study, the data were available from January 1981 to present but recently they were extended to cover the period from January 1951 to December 1980. We used observed data from the KNMI Climate Explorer to assess and correct the ERA5-Land extreme precipitation estimates due to the discrepancy between point station data and grid averaged data. Figure 2 shows monthly averages of rainfall in the region.
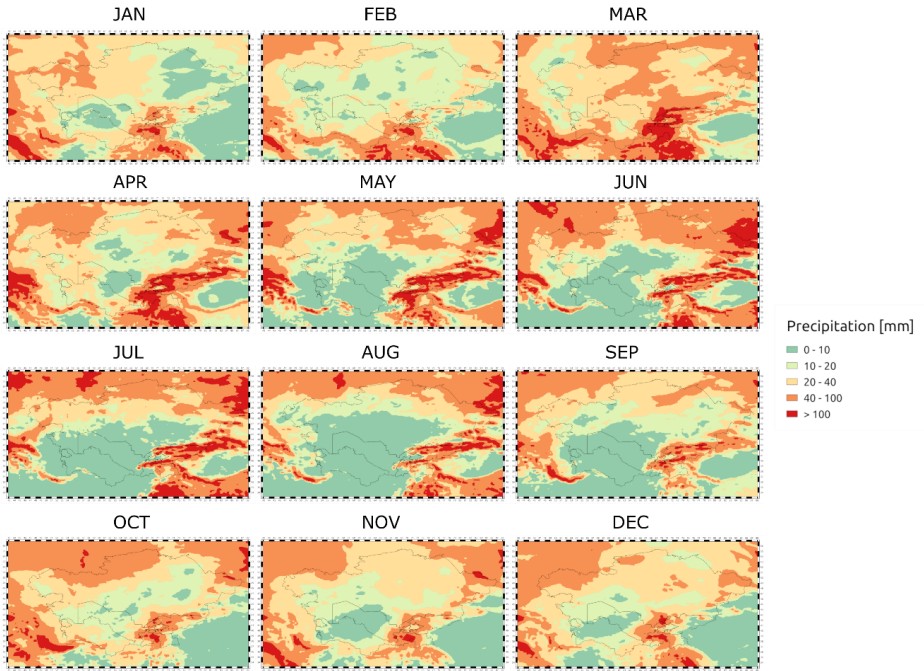

**Figure 2. ERA5-Land monthly average precipitation during the period 1981-2020.**

We also utilized the GlobeLand30 dataset as a source of land use information. GlobeLand30 is a Global Land Cover dataset developed by the National Geomatics Center of China (NGCC), featuring a 30-meter resolution and encompassing 10 distinct land cover classes. This dataset has been made accessible to the United Nations (UN) through the UN-ESCAP Statistics Division and the UN_ESCAP ICT and Disaster Risk and Reduction Division. For the purpose of delineating soil types and analyzing hydraulic properties, the FAO Harmonized World Soil Database was employed. This comprehensive dataset comprises a 30 arc-second raster database encompassing more than 15,000 distinct soil mapping units. It effectively integrates updated soil information from various regional and national sources on a global scale (Nachtergaele et al., 2012).

MERIT DEM (Karnieli et al., 2010) together with MERIT Hydro (Yamazaki et al., 2019) were used as a source of elevation data. MERIT Hydro is a global flow direction map at 3 arc-second resolution (~90 m at the equator) derived from the latest elevation data (MERIT DEM) and water body datasets G1WBM, GSWO (which stand for Global 1 arc-second Water Body Map and Global Surface Water Occurrence, respectively, and were developed by the same authors of MERIT) as well as OpenStreetMap (Coast, 2005). MERIT Hydro, which was produced at the University of Tokyo, uses an innovative algorithm to extract river networks near-automatically by separating actual inland basins from dummy depressions caused by the errors in input elevation data. MERIT Hydro improves on existing global hydrography datasets in terms of spatial coverage and representation of small streams, mainly due to increased availability of high-quality baseline geospatial datasets.

Daily streamflow records from the Global Runoff Data Centre (GRDC) were used as a source of flow records. The Global Reservoir and Dam Database (GRanD), which provides information on dams' location, storage and first year of operation for over 6,862 dams and reservoirs worldwide (The Global Water System Project, 2011) was used to identify dams and lakes.

The strategy for delineating hydraulic defenses in the region incorporates predominantly localized and qualitative data on hydraulic protection gathered from local partners. This information is synergistically combined with the FLOPROS database, which offers established hydraulic protection standards (Scussolini et al., 2016). Additionally, the strategy leverages datasets such as WorldPOP and Landsat HBASE, providing insights into urban areas and population density density (Tatem, 2017; Wang et al., 2017a), thereby enhancing the comprehensiveness of the approach.





Table 1 shows a complete inventory of the global data and how it was used within the model.


**Table 1. Global data inventory.**

| Data Type | Source | Use within the model |
|---|---|---|
| **Digital Elevation Model** | MERIT DEM<br>MERIT Hydro | Input to TOPKAPI, CA2D.<br>Input to derive additional data products |
| **Soil Type** | FAO Harmonized World Soil Database | TOPKAPI parametrization |
| **Land Use** | GlobeLand30 | TOPKAPI and CA2D parametrization. |
| **Observed Discharge records** | Global Runoff Data Centre (GRDC) | TOPKAPI calibration and extreme values distributions |
| **Precipitation** | ERA5-Land: 1981-2020<br>KNMI Climate Explorer | Input to TOPKAPI and CA2D pluvial simulations |
| **Temperature** | ERA5-Land: 1981-2020 | Input to TOPKAPI |
| **Reservoirs and Dams** | Global Reservoir and Dam Database (GRanD) | Extreme Values Analysis and Regionalization |
| **Hydraulic defenses** | FLOPROS database<br>WorldPOP<br>HBASE | Defended hazard maps |

The impact of climate change was accounted for based on climate projections from the Coordinated Regional Climate Downscaling Experiment (CORDEX) project database (Giorgi et al., 2009). In particular, the model results of the regional climate model RegCM4.3.5 driven by the MPI-ESM-MR global model in the Central Asia

region (Ozturk et al., 2017), were used, under the scenario RCP4.5 (van Vuuren et al., 2011).

### 3.2 Local datasets

Within the SFRARR project, multiple workshops and meetings with local stakeholders and experts were held. In particular, eight capacity building workshops devoted to the different risk assessment components, namely five country-based workshops on exposure assessment and three regional thematic workshops on hazard, vulnerability

and risk modelling. This activity was carried out in close collaboration with local experts and representatives from all five countries. The workshops provided participants an opportunity to learn about international best practice and latest methodologies related to natural risks assessment. These workshops allowed sharing knowledge with local experts and provided an opportunity for emergence and inclusion of a greater amount of locally collected information into the analysis.

Obtaining daily discharge and hydraulic protection data from local sources proved complex due to variability in data quality and form. Compiling comprehensive hydraulic protection data at the country level was hindered by its highly classified and confidential nature, posing challenges for acquisition.

Table 2. shows a complete inventory of the data requested/obtained from the local experts and stakeholders.

Although the number of available flow gauges might appear limited given the extensive region, it is crucial to

recognize that their spatial distribution effectively encompasses densely populated areas where the majority of exposed assets are located. Figure 3 and Figure 4 show the gauging stations locations and the populated areas (in blue, population density > 1/km$^2$) including both local and global datasets.





**Table 2. Local data inventory.**

| Country | Daily Discharge | Annual maximum discharge | Hydraulic Protection | Reservoirs |
|---|---|---|---|---|
| Kazakhstan | 7 stations (records of variable lengths between 2001 and 2015) | 120 stations (records of variable lengths between 1910 and 2018) | Location and length of some riverbank hydraulic structures on the Sir Darya River | Volume and year of construction of main reservoirs |
| Kyrgyz Republic | No data obtained because cost was too high compared to the benefit | 65 stations (records of variable length between 1930 and 2018). Some of these data were purchased from KyrgyzHydroMet | Record of hydraulic protection works from 2018 at the Oblast level | Data on reservoirs' volume and construction year for 5 main reservoirs |
| Tajikistan | No data obtained because cost was too high compared to the benefit | 14 stations (variable lengths), purchased from TajykHydroMet | No data | Volume and construction year for 13 reservoirs |
| Uzbekistan | 2 stations (2015-2019) | 46 stations (2005-2019) | No data | Volume and year of construction of main reservoirs |
| Turkmenistan | 6 stations (2015-2020) | 11 stations, variable record between 1936-2020 (monthly maxima available) | No data | Volume and year of construction of main reservoirs |

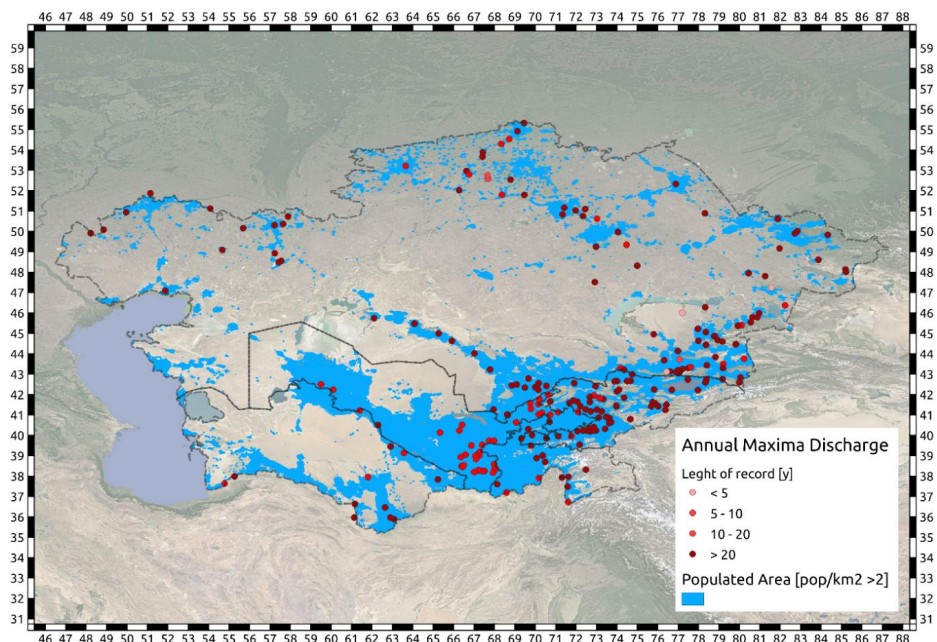

**Figure 3. Annual maxima flow gauges coverage on populated areas, provided by local Institutes**


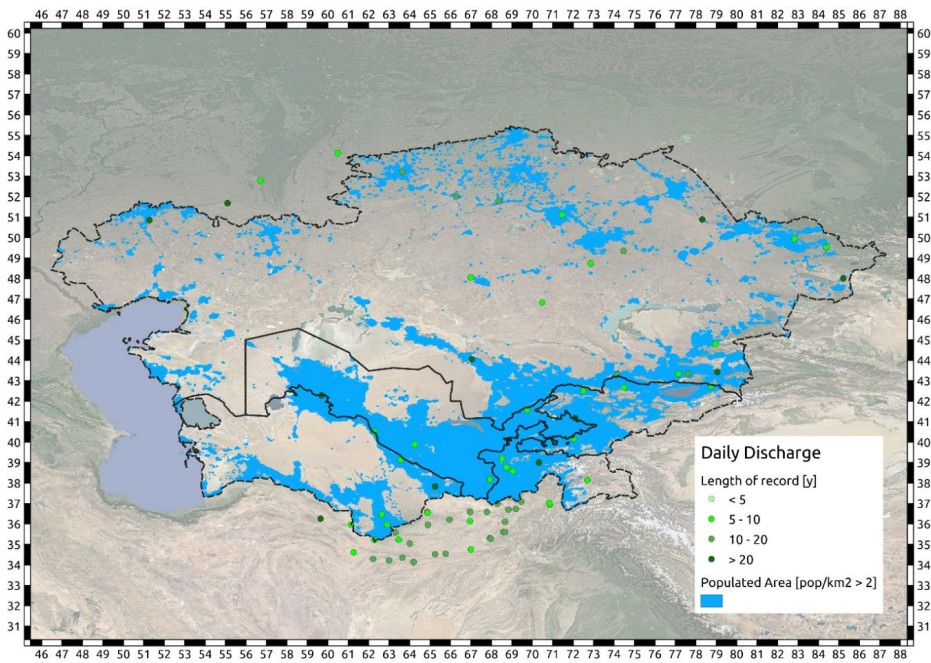

**Figure 4. Daily flow gauges coverage on populated areas, provided by local Institutes (inside of the country borders) and GRDC (inside and outside of the Countries' borders)**

Information about the characteristics of the building stock of relevance for their vulnerability to flood water was collected from local sources and from the literature. These characteristics include number of floors, presence of a basement, level of the ground floor above the street level, type of building (apartment, detached, semi-detached), etc. The number of floors distribution in various countries was mainly derived from (Pittore et al., 2020) and (Wieland et al., 2015), which established floor number ranges for different building categories through local surveys. Additionally, sources such as (Pittore et al., 2011) and (The World Bank, 2017) were consulted for the Kyrgyz Republic, while 2538 surveyed buildings in Dushanbe (Tajikistan) provided insights. On-site collection of unit costs for building component maintenance, removal, and replacement was facilitated by local advisors and engineers, drawing from interactions with professionals involved in building design and pricing, as well as engineering manuals and real estate catalogs (such as ENiR - Uniform norms and prices for construction, installation, and repair works).

### 3.3 Models

As per the flood hazard assessment of the five Central Asia countries, the numerical modelling toolset is composed of two elements: the hydrological model (TOPKAPI-X) and the flood hydraulic model (CA2D).

The TOPKAPI (TOPographic Kinematic APproximation and Integration) model is a fully-distributed physically-based hydrological model that can provide high resolution information on the hydrological state of a catchment (Ciarapica and Todini, 2002). The TOPKAPI-X model is an advanced version of the original TOPAKPI model that includes an additional soil layer for assessing subsurface flow, an improved snow melting and accumulation module that considers terrain aspect and latitude and a groundwater component to model the aquifer flow. The TOPKAPI-X model requires as input both precipitation and temperature meteorological data, plus a description of the soil characteristics that can be derived from the land use (to derive crop factors and surface roughness) and soil type maps (to derive soil permeability and depth).

CA2D (Dottori and Todini, 2011) is a full physically-based flood model specifically designed for high-performance computing applications, based on the cellular automata (CA) approach and the diffusive wave equations, to simulate flood inundation events involving wide areas. A Cellular Automata (CA) is a cell-centered,





finite volume scheme for the description of physical dynamic systems regulated only by local laws. In this scheme, each cell assumes a certain state (or property), which evolves in time according to specified fixed rules. In CA2D case, the momentum equation is solved for each time step, computing volume exchanges between grid cells along the cell's borders. Volumes of each cell are successively updated using flow volume conservative equations. The model is based on the state-of-the-art of large-scale hydraulic modelling and has been tested extensively on several case studies. The CA2D model has an internal preprocessor that allows the user to provide as input only the Digital Elevation Model and the surface roughness map. The network (comprising nodes and links) is automatically generated, and specific conditions (such as flood protections) can be included, where present. In addition, input meteorological data must be provided in the form of hydrographs at specific points and/or of rainfall maps. We ran the model using the semi-inertial formulation of the momentum equation, which was developed for the LISFLOOD-FP model (Bates et al., 2010). This approach allows for high resolution simulations at a significantly reduced computational effort, making it possible to run hydraulic simulations at the continental and global scale (Dottori and Todini, 2011).

Flood vulnerability for buildings was assessed using a component-based flood vulnerability model, called INSYDE (Dottori et al., 2016). This model account for different measures of the event intensity (water depth, but also flow velocity, flood duration, sediment load, water quality, etc.) and different components of the building (structural, non-structural, finishing, doors/windows, systems, basement, etc.) to derive a large set of curves for each component of the damage. These curves are then combined depending on the characteristics of the building categories. Local knowledge was key in the construction of vulnerability curves for buildings, in terms of defining unit costs of the components, archetype buildings, materials, etc. The infrastructure vulnerability (e.g., roads, power plants, airports) was taken from the Global Flood Depth-Damage dataset developed by the European Union's Joint Research Centre (Dottori et al., 2018; Huizinga et al., 2017).

An exposure database for the region (Scaini et al., 2023a, b), which includes residential and non-residential buildings, transportation infrastructure and crops, was developed by assembling available global and regional datasets with country-based information provided by local authorities and research groups, including reconstruction costs.

The risk assessment has been performed using the CAPRA platform (www.ecapra.org), which is an open-source and free platform for multi-hazard probabilistic and deterministic risk assessment, that has been developed with the initial financial support of the World Bank, the Inter-American Development Bank and the UNISDR (Reinoso et al., 2018). The CAPRA Platform, which allows multi-peril assessment (using the probabilistic methodologies described in this manuscript) uses the geographical information (for the exposure and hazard components) and produces economic losses aligned with the risk metrics typically employed in the insurance industry, besides. Moreover, it produces GIS-compatible geospatial data layers with metadata, describing estimated loss per administrative unit, as well as identifying the location of key industrial sites, critical and supply infrastructure and the corresponding hazard intensity values at those locations, either in raster or vector formats.

## 4 Methodology

### 4.1 Flood hazard assessment

In this study, the flood hazard of the five Central Asian was assessed by means of a physically based numerical modelling toolset and a stochastic catalogue of flood footprints (Figure 5).

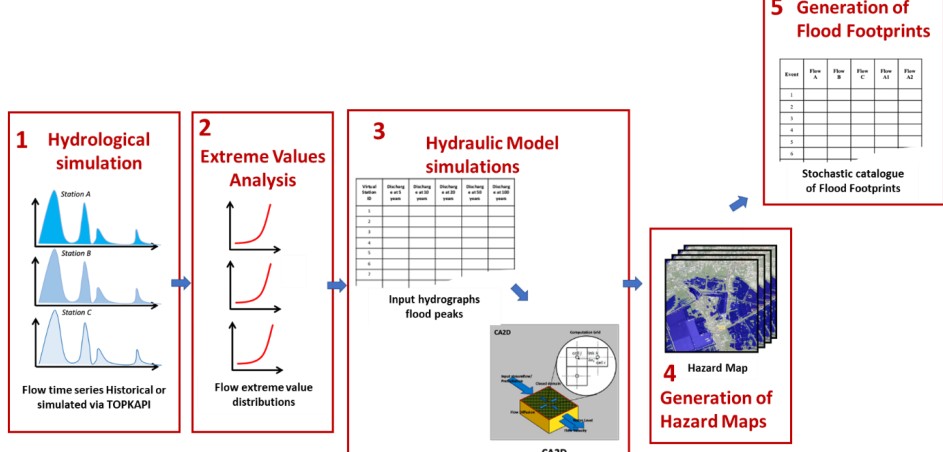

**Figure 5. Flood hazard assessment: schematic representation of methodology.**

The modelling steps followed in this study were:

1. ERA5-Land hourly precipitation and temperature on a 0.1×0.1° regular grid were processed and used to drive calibrated TOPKAPI-X set-ups (one for each catchment in the region, including catchments partially overlaying neighbouring countries such as China, Afghanistan and Russia). The output was a set of 40-years long hourly discharge values at numerous river sections covering the whole drainage network in the region.

2. For each river section, an extreme value analysis was carried out by fitting a Generalised Extreme Value distribution (Jenkinson, 1955) over the 40 annual maxima derived with TOPKAPI-X. Given the shortness of this series, the uncertainties in the input data and the alterations to flow regime caused by lakes and dams, a correction procedure was established to incorporate flow records, where available. This procedure corrects the resulting extreme value distributions with the objective of maximising the similarity between observed and modelled extreme flows. The output of this step is composed of several discharge values at fixed exceedance probabilities (1-in-5, 10, 20, 50, 100, 200, 500, 1000 years floods) for each river section.

3. At each river section, the CA2D model was used to simulate the flood propagation of the river discharges generated at step 2), producing reach-specific water depth footprints for each of the fixed exceedance probability levels.

4. The single reach-specific footprints were joined to obtain hazard maps for 1-in-5, 10, 20, 50, 100, 200, 500, 1000 years exceedance probabilities.

5. A stochastic catalogue of events was generated by assessing the historical correlation among flows of all the river reaches and extracting randomly reach-specific footprints from the hazard maps according to the cross-correlation derived from historical flow series.

More information about these five steps is provided below.

The TOPKAPI-X model was run on a regular 1×1 km grid for all catchments in the region, based on a resampled digital elevation model consistent with MERIT-Hydro in terms of flow direction and river network. Soil type and land use maps were also resampled to match the same grid. The model was run on an hourly time-step using hourly ERA5-Land precipitation and temperature from January 1981 to December 2020. The model simulations were initiated with average soil saturation and river depth conditions and the first year of simulation was used as warm-up period to reach realistic soil moisture conditions. Therefore, the year 1981 was not considered for calibration purposes nor for the extreme value analysis. The main model output consisted of hourly simulated discharges at several locations of the river network across the entire region. The model simulations corresponding to locations where observations were available have been used to perform a trial-and-error calibration that could reasonably reproduce the overall behaviour of each catchment.





The extreme value analysis and regionalisation process was based on fitting the General Extreme Value (GEV) distribution for several locations along the drainage network to derive the peak flows with different return periods. GEV is a standard tool for modeling flood peaks using annual maximum series (Morrison and Smith, 2002; Rosbjerg and Madsen, 1995). Simulated flow annual maxima were used to derive a GEV distribution for a large number of river sections all over the river network. Where observed flow records were available, the GEV distribution was also fitted on the observed flow annual maxima, and the resulting distribution compared with the distribution derived from simulated flow values with the objective of evaluating the model error on the extreme values. We observed that the largest hydrological model discrepancies occur on the main stem of large rivers, because of the impact of large reservoirs and floodplains. Therefore, an adjustment of the simulated flows was implemented by computing the ratio between observed and simulated mean annual maximum flows and then multiplying the simulated flow by such a ratio. In other terms, the simulated annual maximum flows were increased or decreased by a coefficient based on the mean bias between observed and simulated mean annual maximum flows. Where observed data were not available, the adjustment coefficient was computed by using an adjustment coefficient from an associated station selected based on the proximity of the location and flow accumulated area. This procedure yielded a very good fit between observed and simulated extreme values of flow and allowed extrapolating the adjustment to ungauged river sections. This adjustment was particularly useful in floodplains and downstream dams, which are features that are particularly difficult to reproduce with a hydrological model. With this procedure we obtained estimates of the extreme flow value distribution for 78,000 river sections having more than 100 km$^2$ drainage area, from which peak flows were extracted at several levels of likelihood (1-in-5, 10, 20, 50, 100, 200, 500, 1000 years).

The CA2D model was run at a 0.00083° (~90 m) spatial resolution. The simulation time-step is dynamic, and it varies between 0.01 and 15 seconds, with the maximum allowable time step defined by the Courant–Friedrichs–Lewy or CFL condition (Courant et al., 1928), which is commonly adopted to preserve stability in computational fluid dynamics models. For every one of the 78,000 river reaches, 8 simulations were carried out using the 1-in-5, 10, 20, 50, 100, 200, 500 and 1000 flows resulting from the extreme value analysis as boundary conditions. The MERIT-Hydro model was used as a source of elevation data and GlobeLand30 was used to derive roughness values from land use classes. For each river section, we built a flood hydrograph to estimate the time of concentration, i.e., the time needed for the water to flow from the most remote point in a watershed to the watershed outlet (Giandotti, 1934). We assumed a triangular hydrograph reaching the flood peak at 2/3 of the concentration time and going back to zero at twice the concentration time. We assume the bankfull discharge as the discharge at 2-year return period.

The role of the defensive protections is crucial in reducing fluvial flood hazard. However, availability of precise data regarding flood protection levels is very limited, as discussed earlier. To circumvent this problem, we developed a strategy to derive the hydraulic protection level of the region based on the correlation between the level of protection and the population density at any given location along the river. The estimated correlation was then linearly scaled using the area protection standards provided by the FLOPROS database (Scussolini et al., 2016). To apply this strategy, we first identified urban agglomerates and calculated the corresponding maximum population density using the HBASE (Wang et al., 2017b) and WorldPop (Tatem, 2017) datasets. The former indicates the extension of urban areas, while the latter provides population density over 1 km$^2$. We identified the river portions afferent to each urban agglomerate by assuming that each river reach, within a variable distance from the agglomerate whose amount depends on the accumulated area, has the same urban level of protection. Finally, we assigned a level of protection to each river tract afferent to the agglomerate, based on the relationship previously estimated with the population density and the area's FLOPROS protection standards. When possible, we integrated into these estimates the localized and mainly qualitative data obtained from the local partners. This was the case, for example, for the level of protection of the two main Kazakh cities (Almaty and Astana), for which we utilized geospatial information provided by local stakeholders.

The results of the flood model simulations were the hazard maps, i.e., water depth maps at fixed return periods. While hazard maps provide the depth of inundation that can occur at a given location with a certain annual probability (or, conversely, with a certain return period), they are unable to describe the likelihood of concurrent flooding across multiple sites. This caveat limits their capability of assessing risk over the full range of plausible scenarios, including the most extreme ones which are those of most concerns to stakeholders. For this purpose, risk assessment models routinely use stochastic catalogues of events, i.e., datasets of synthetic event intensity

footprints. This procedure is typically used for rainfall events (Salazar et al., 2009; Francés et al., 2011), tropical cyclones (Bloemendaal et al., 2020), drought (Guillod et al., 2018) and other perils. In this study, a flood depth hazard catalogue serves this purpose, by providing a stochastic ensemble of 10,000 years of hypothetical floods that may occur in the region, with related annual frequency of occurrence. To ensure spatial coherence in the stochastic catalogue, the spatial correlation of the river flow at each gauge/station is determined by computing a cross correlation matrix based on all the available (observed/simulated) flow time series. The methodology followed to produce a stochastic catalogue consists of the following steps:

1. Clustering: river sections are grouped into clusters under the assumption that flows at river sections in the same cluster are highly correlated random variables. The amount of correlation depends on historical simulated flows, location of the stations and accumulated areas of the stations.
2. Cluster activation probability: the annual probability of activation of a cluster is computed, where "activation" is defined here as an instance when at least one river section in a given cluster exceeds the 5-year flow. This probability is based on the activation of clusters in the historical simulated flows.
3. Activation of river section within a cluster: the average number of active river sections for a given year and its standard deviation are computed. A station is defined to be active when the flow at that location exceeds the 5-year flow. These values are based on the activation of clusters in the historical simulated flows.
4. Generation of the stochastic catalogue: based on all the analysis above (clusters, annual activation probability, average number of active stations) and on the hazard curves at each section, a stochastic catalogue is produced, with equivalent duration of 10,000 years. Every year consists of an annual flood footprint, i.e., a map where each pixel represents the maximum water depth during a given year.

The impact of climate change on flood hazard was accounted for by estimating change factors to the ERA5-Land precipitation and temperature based on the probability density function (pdf) comparison of the current climate and 1971-2100 projection. We used a probability density function matching technique to modify the distributions of the current ERA5-Land variables (Lafon et al., 2013). To derive the hazard maps for the time horizon 2080, the entire modeling chain composed of hydrological modeling (TOPKAPI-X), extreme values analysis and hydraulic modeling (CA2D) was fed with the modified ERA5-Land derived meteorological input data (precipitation and temperature). Although we are fully cognizant that flood hazard and risk estimates under scenarios of climate change are affected by a very large uncertainty (Bubeck et al., 2011), it is of paramount importance that the effects of climate change be considered into any disaster risk management strategy to ensure robust planning going forward.

**4.2 Flood risk assessment**

As stated earlier, the flood risk of the five Central Asian countries was assessed by means of the CAPRA risk assessment software (Reinoso, Eduardo, Ordaz, Mario, Cardona, Omar-Dario, Bernal, Gabriel A, Contreras, 2018) according to the methodology displayed in Figure 6.

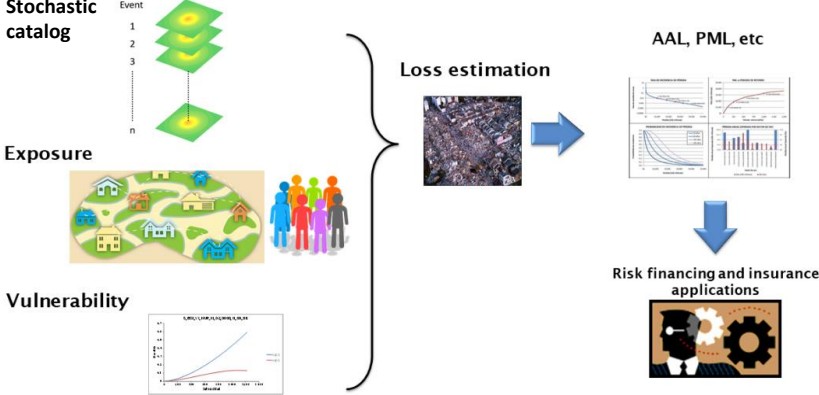

**Figure 6. Flood risk assessment: schematic representation of methodology.**



The loss estimation module allows estimating both the economic losses for the assets in the exposure datasets and corresponding human losses for each of the possible future events in the stochastic catalogue. Economic losses for each exposed asset are determined by combining the flood depth distribution at the site with the corresponding damage function. This yields a distribution of mean damage ratios (repair cost divided by asset replacement cost) for each asset. Scaling this distribution by the total asset value generates the loss distribution caused by a flood event. Summing these losses for all exposed assets provides the total loss for the event.

The flood risk model introduced is designed to furnish necessary loss metrics for formulating risk reduction strategies, including insurance product design. Key outcomes of the long-term flood risk assessment are the year loss tables (YLTs) for each of the five analyzed countries. These YLTs detail economic loss expectations, uncertainty, annual occurrence frequency, and a timestamp (ranging from year 1 to year 10,000) for each event in the stochastic set. Using this simulated data, one can estimate either the aggregate yearly occurrence probability of losses exceeding a threshold or the annual probability of the maximum event loss surpassing a threshold. The YLTs enable the derivation of a loss exceedance curve (LEC), which encapsulates loss occurrence characteristics and informs disaster risk management activities, such as regional and national risk financing and insurance development. Additionally, the model yields estimate of key risk measures like average annual loss (AAL) and probable maximum loss (PML), commonly used for risk communication.

The flood risk model presented here is designed to provide all the loss metrics needed to devise risk mitigation strategies, including the design of an insurance product. Key outcomes of the long-term flood risk assessment are the year loss tables (YLTs) for each of the five analyzed countries. These YLTs detail the expected value and the corresponding uncertainty of the economic loss, together with its annual frequency of occurrence, and a timestamp (ranging from year 1 to year 10,000) for each event in the stochastic set. The YLTs can then be used to derive the loss exceedance curve (LEC), also known as the Exceedance Probability (EP) curve (e.g., Mitchell-Wallace et al., 2017), which encapsulates loss occurrence characteristics and informs disaster risk management activities, such as regional and national risk financing and insurance development. Additionally, the model yields estimate of key risk measures like average annual loss (AAL) and probable maximum loss (PML), commonly used for risk communication.

Risk model calibration and validation are typically carried out by comparing modelled and observed loss estimates for historical events, adjusting some of the model parameters or components to improve the goodness-of-fit. Observed loss estimates are usually obtained from post-event assessment reports and surveys. However, limited and uncertain historical flood loss data in Central Asia hinder this calibration, especially as detailed flood depth maps are rare. Such maps enable the reduction of the uncertainty in the estimation the water depth estimates, so that any discrepancy between observed and modelled losses can be more clearly attributed to the vulnerability or the exposure components of the risk model. In Central Asia unfortunately only one historical flood event has these characteristics (flood depth map and post-event loss assessment), namely the 2005 Hamadoni flood in Tajikistan (M.S. Saidov, Yu.N. Pilguy and L.V. Davlyatshoeva, 2006). This scarcity of loss data limits the efficacy of the flood risk model calibration effort.

Bearing in mind these data availability limitations and the objective of the present risk assessment (which is to estimate the underlying, long-term average flood risk), the model calibration was carried out as follows:

1. A list of historical events and reported losses was collected;
2. The districts/regions affected by the historical floods were identified;
3. The risk model was run using the stochastic catalog of flood footprints as input, for all the district/regions previously identified;
4. The exceedance probability curves of all selected district/regions were calculated based on the results of the simulations with the stochastic catalogue;
5. Based on the resulting exceedance probability curves, the return periods of the historical losses were computed (historical losses and district/region losses are comparable under the assumption that reported events are usually large floods that either affect the whole district/region or represent economic losses that are significant at the scale of the whole district/region);
6. The resulting return periods were critically analyzed under the following assumptions:
    a. Reported events are typically large events that make the news, and therefore are relatively rare. It is expected that their return period is at least 5 years.
    b. It is relatively unlikely that a reported flood event has a return period of more than 500-1,000 years.


       c.    If a region has more than one reported event, it is highly unlikely that all events have return periods longer than 100 years.

445        d.    In general, it is expected that most reported floods have a return period between 5 and 100 years, with very few outliers.

    7.   If some of the above criteria were not met, the vulnerability curves were adjusted to increase or decrease the losses and obtain a better adjustment to the criteria.

The rational of this methodology is that, instead of providing direct comparisons between observed and reported
losses (not possible given the lack of available data), the calibration process tries to demonstrate that the model is providing risk estimates that are in line with what has been observed in the past 20 years in terms of frequency of the events and severity of the economic losses. Given the objective and the limitations described above, this seems to be the most tenable strategy that both exploits the, albeit rare, data available and provides sensible loss estimates.

**5 Results**

**5.1 Calibration and validation**

**5.1.1 Hydrology/flood model**

The TOPKAPI-X model was calibrated through a trial-and-error procedure adapting the initial model parameters in order to match the available observed discharge. We mainly based the model calibration on the historical daily
data, but also used the annual maxima for the areas where daily data were not available. The calibration process focused mainly on robustly reproducing the flow peaks. This is because the hydrological simulations aim at estimating the extreme discharge value distributions at river sections across the region that are used to derive the flood footprint at different return periods via the hydraulic model. Since a physically based model better reproduces the flow peaks if all the other hydrological components are well represented, we made sure that the
main hydrological processes were also correctly reproduced, with particular attention to the snow accumulation/melting component, which is the driver of most of the floods in this region. The calibration has been performed independently for each catchment where historical data were available. Given the distributed and physically based nature of the TOPKAPI-X hydrological model, the calibration process was not based on an automatic procedure but on the use of reasonable values of the physical parameters. This procedure allows the
identification of model parameters values that provide reasonable outputs across the entire catchments, avoiding under- or over-fitting any of the available historical records. During the extreme value analysis calibration process, the main limitations of the hydrological model application were considered:

- Absence of lake/reservoirs modeling. The influence of lakes and reservoirs was taken into account by allowing an overestimation of the flood peaks and underestimation of the low flows for for downstream
stations of primary reservoirs. These reservoirs effectively attenuate flood waves by storing water during peak events and releasing it during drier periods. This behavior was not explicitly modeled due to lack of information on the reservoir management strategies. Hence, it was anticipated that the model would not precisely replicate observed hydrographs for stations located downstream of reservoirs (e.g., the Turkmenabat, Kerki (former Atamurat), Semiyarskoje and Chirchik stations in Figure 7).
- Inability to reproduce the effect of alluvial plains, which similarly to reservoirs alter flood waves by flattening peak flows. To address this, the same approach was adopted as with reservoirs, namely we allowed the overestimation of the peaks and the underestimation of recession curves (e.g, evident at stations like the Kushum and Gram in Figure 7) for stations located within alluvial plains.

Figure 7 shows observed (blue line) and TOPKAPI-X simulated (red line) discharge for a series of relevant
catchments and locations, along with temperature and precipitation records. As expected, the presence of reservoirs and lakes has a clear impact on the simulation performance: the sites of Turkmenabat and Kerki (former Atamurat) on the Amu Darya River for example, are located downstream a series of reservoirs, which are not modeled in TOPKAPI-X because of lack of data. This causes the model to overestimate discharge in spring/summer when the reservoirs retain part of the snowmelt, and underestimate discharge in the rest of the
year, as the reservoirs is slowly released (Figure 7). A similar behavior is observed at the Semiyarskoje and





Chirchik sites on the Irtysh and the Syr Darya rivers, respectively. The two stations are located downstream of a lake (Zajsan lake's area is 1.810 km$^2$, Charvak lake's is 40 km$^2$), which explains the shift and delay in streamflow which is not equally observed in the respective catchment's upstream sites. The station at Kushum (Ural river) is located within an alluvial plain, whose lamination effect cannot be entirely reproduced by the TOPKAPI-X model

and we allowed the model to overestimate the peaks and underestimate the recession curves. At stations Orenburg (Ural river) and Ulba Perevalochnaya (Amu Darya river), where these limitations do not apply, the model effectively reproduces discharge patterns throughout the entire period.

From an inspection of Figure 8, which shows a comparison of observed and simulated maximum flows for a series of selected sites, it is apparent that the TOPKAPI-X model exhibits a favorable correlation and minimal bias in

simulating annual maximum discharge for a majority of the locations.

As stated above, the acknowledged limitations of hydrological modeling across such a large area (namely effects of lakes/reservoirs and alluvial plains) were addressed during the extreme value analysis, resulting in adjustments that improved alignment with available observations. This refinement facilitated the derivation of realistic input hydrographs for the hydraulic modeling component.


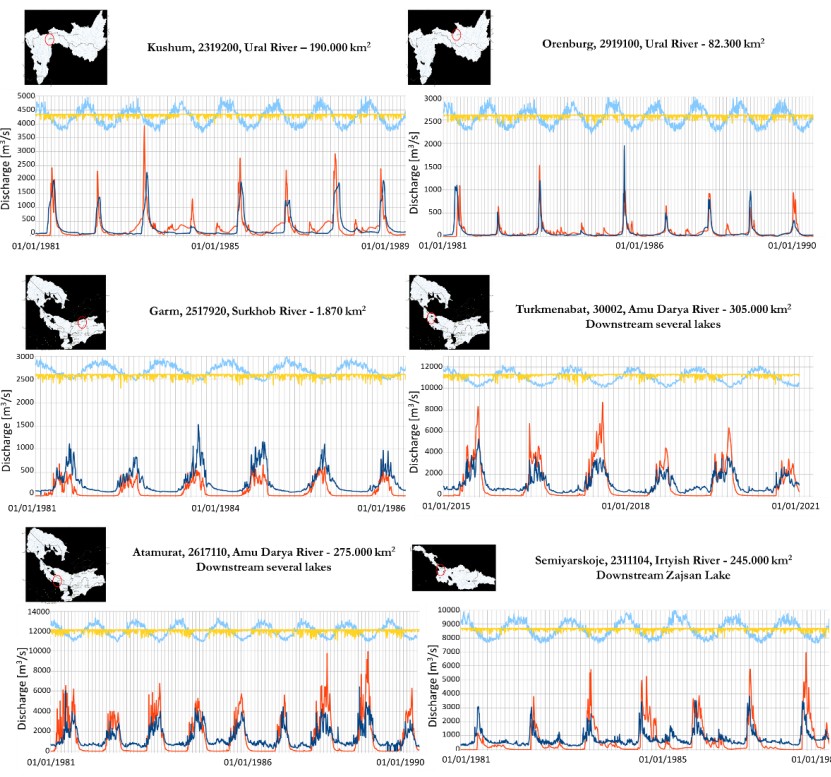


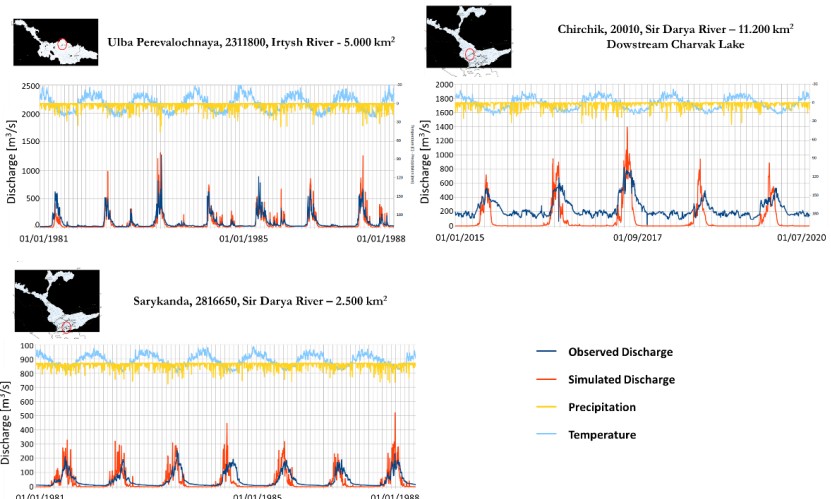

**Figure 7. Example of observed (blue line) versus TOPKAPI simulated (red line) discharge computed at various locations for selected river sections. The effect of dams (e.g., Turkmenabat and Kerki (former Atamurat)) and floodplains (e.g., Kushum) can be observed but the flow peaks are, in general, well reproduced.**


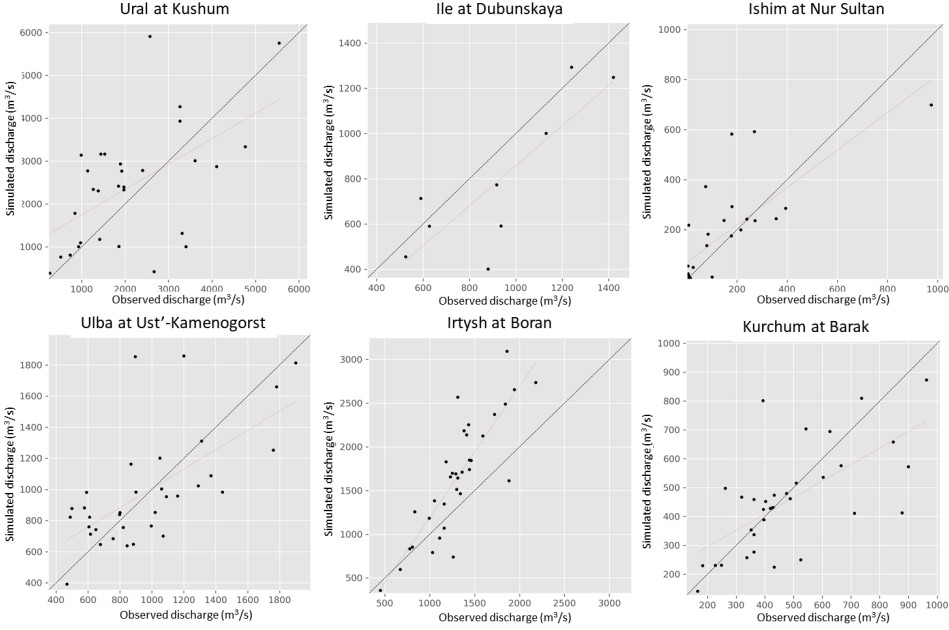

**Figure 8. Observed versus TOPKAPI simulated maximum annual discharge for selected sites. The black lines indicate the 1:1 line, the red lines indicate the trendline fitted on the scatterplot.**

Trend lines in Figure 8 show that the simulated annual maxima for some stations are in some cases biased compared to the observed maxima. This issue has been dealt with in the extreme value analysis through an adjustment of the simulated annual maxima, as described in the Flood Hazard Assessment Section.

### 5.1.2 Risk

Although all the components of the risk assessment developed in the project (hazard, exposure, and vulnerability) have been separately validated against observations to the extent possible, it is good practice to calibrate and validate the risk model as a whole. This further calibration step, when needed, often adjusts the exposure and vulnerability module to ensure a better agreement among historical observations of economic losses and modelled losses.

Given the data limitation, it was decided to reduce the number of calibration parameters to a minimum. Hence, all the vulnerability curves were increased or decreased by the same amount, i.e., no differential calibration was carried out on vulnerability curves of different exposure classes or different countries. Furthermore, only the residential building vulnerability curves were calibrated, since residential buildings account for the majority of the exposed value. Infrastructure and crop vulnerability functions were left unadjusted, as no data were available to justify a specific calibration of such curves. The results of the calibration for the flood vulnerability curves

yielded an increase of the overall vulnerability of 20%.

Human vulnerability curves were calibrated based on national-scale statistics of fatalities caused by floods. Vulnerability functions were adjusted so that the average number of fatalities per year provided by the model was similar to the values obtained from the official statistics.

The reported monetary and human losses were collected with the support of the local partners. Although affected

by large uncertainties, these are the only datasets available for model calibration, as presented below.

*Historical flood (Hamadoni 2005 flood)*

Figure 9 shows the simulated flood footprint of the June-July 2005 Hamadoni flood in Tajikistan while Figure 10 presents the comparison of the modelled and reported economic losses and fatalities. The average value is indicated in green while, for losses only, the upper and lower limits of the values reported by different sources are

shown by the black whiskers. For fatalities, only one value was reported.

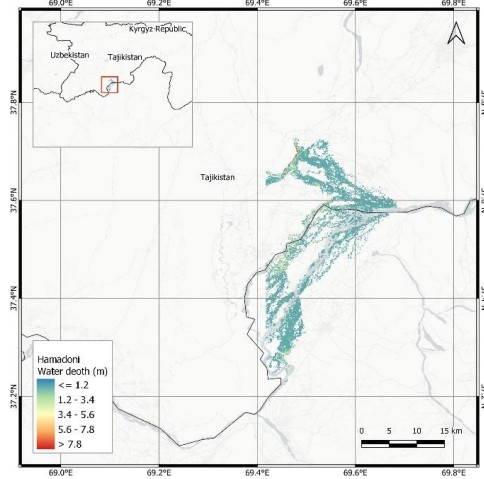

**Figure 9. Simulated flood footprint of the 2005 Hamadoni event in Tajikistan**



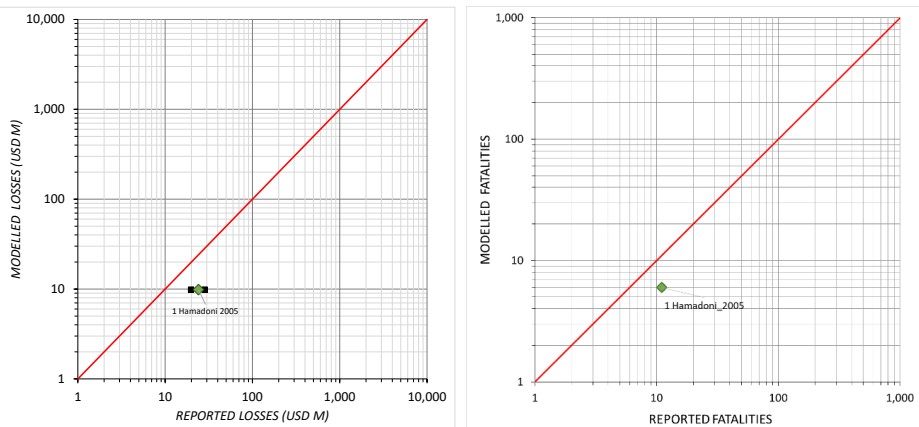

**Figure 10. Comparison of modelled economic losses (left) and fatalities (right) for the 2005 Hamadoni flood in Tajikistan**

In this case, the average value of the reported economic losses for the Hamadoni 2005 flood are higher than the modeled losses by a factor of approximately 2.5. The observed losses are often larger than the modelled ones since the observations likely include indirect losses, such as those caused by business interruption due to road closures,
which are not considered in this model. Furthermore, the trending of the losses to current monetary values is likely to have introduced a considerable uncertainty. As an illustrative example of the effects of trending on the variability of the observed losses in current money, the original reported loss values range between 7 and 10 M (2005) USD, whereas the trended reported losses are between 19 and 28 M USD.

The 2005 Hamadoni flood is the only event for which reported losses, flood footprints and river flow time series
are available. For this reason, the event could be reconstructed with the model and a direct comparison between event-only modelled and observed losses was carried out. Nevertheless, the whole model cannot be calibrated based on a single modelled-observed loss comparison, as this would not be a robust calibration strategy. Several uncertainties affect both the observation (post-event loss assessment methodology, trending of 2005 losses to current monetary values, inclusion of indirect losses which are not included in the model, etc.) and the model
results (digital elevation model resolution, dyke breach mechanism, etc.). For this reason, the calibration was based on the estimation of the return period for all historical events according to the oblast-scale exceedance probability curve. The 2005 Hamadoni flood was used as a further sanity check to validate the order of magnitude of the losses.

*Flood results validation using the EP loss curves*

To complement the Hamadoni-flood-centric calibration procedure above, we also compared the EP loss curves for those Oblasts where partial historical loss data were available for some events (Table 3). This procedure consisted of verifying that the reported losses for the 8 historical floods were coherent with the results, by Oblast, in terms of the EP curves (i.e., reviewing that there were no systematic under- or over-estimation of losses). Figure 11 shows one of the results of these comparisons.

**Table 3. Reported economic losses for flood events and identification of the oblast where it occurred**

| Event | Country | Oblast | Reported loss (US M) |
|---|---|---|---|
| Flood_TJK_2010-5-6 | Tajikistan | Khatlon | $307.30 |
| Flood_TJK_2006-4-21 | Tajikistan | Khatlon | $264.68 |
| Flood_TJK_2005-7-23 | Tajikistan | Khatlon | $156.06 |
| Flood_TJK_2021-5-7 | Tajikistan | Khatlon | $8.40 |
| Flood_TJK_2014-4-22 | Tajikistan | Khatlon | $3.04 |
| Flood_KAZ_2010-2- | Kazakhstan | Almaty | $73.44 |
| Flood_KGZ_2012-4-23 | Kyrgyzstan | Osh & Batken | $24.42 |
| Flood_KGZ_2005-6-10 | Kyrgyzstan | Osh & Batken | $8.03 |
|---|---|---|---|

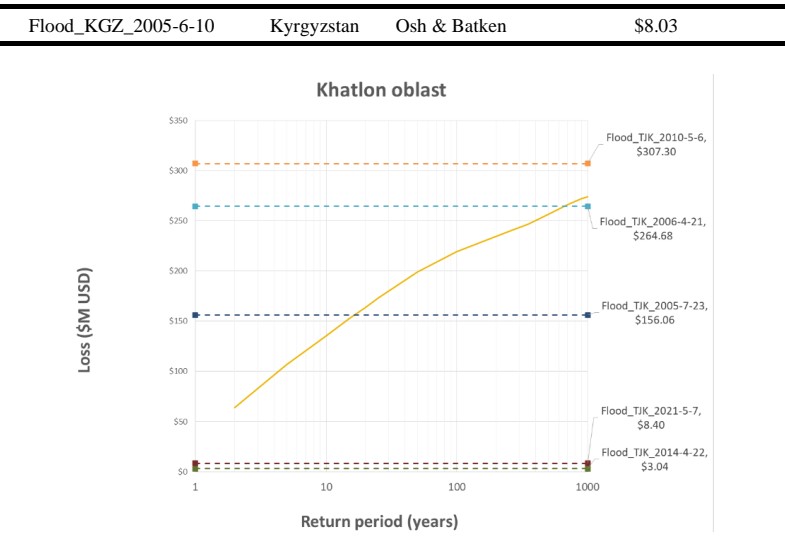

**Figure 11. EP curve and reported economic losses for the Khatlon Oblast in Tajikistan**

When comparing the reported losses with the EP curves (in terms of return periods), we observe that for the case
of the Khatlon Oblast in Tajikistan the April 2006 flood is associated with a return period of approximately 650 years, the July 2005 flood is associated with a return period of approximately 20 years., whereas floods with lower reported losses, such as the April 2014, and May 2021 have a return period of approximately two years. The May 2010 flood corresponds to the event with the largest reported losses, and as per the EP curve calculated for this Oblast, has a return period longer than 1000 years.

Since the Khatlon Oblast has experienced an exceptional number of reported events in the past 20 years, which is uncommon in the rest of the regions, it is reasonable to assume some of the reported events are associated to such short return periods. Furthermore, note that, strictly speaking the comparison between Oblast-wide losses and event-specific losses for the purpose of assessing the reasonability of the associated return periods is not correct. Small events only affect a portion of the Oblast and other events might have happened during the same year.

Therefore, the yearly Oblast losses are, intuitively, larger than event-specific losses that may have occurred in that year. Hence, we expect that small, localized events are associated to short Oblast-scale loss return periods. On the other side of the spectrum, the large May 2010 event in the Khatlon province appears to be out of the limits of the exceedance probability curve. However, it must be noted that there is a large discrepancy among the different data sources in the reported losses for this event: SwissRe reported a loss of around 200 M USD, whereas AON around

5 M USD. The overestimation of observed losses by one of the sources (or perhaps the inclusion of losses by landslides and mudslides, which are not included in this model) might be the cause of the very long estimated return period.

In any case, referring to the event above, the chance that a 10,000yr loss or larger has been observed in 20 years is very small. This rare event can be explained by the discrepancy in the reported losses but, in general, such

extremely large losses associated with very long return periods are not tenable. This is the reason why we calibrated the model to eliminate such cases After calibration, the reported event loss values have plausible return periods when compared to the modeled losses from the subnational EP curves.

### 5.2 Flood hazard

#### 5.2.1 Hazard maps

Given the premises described in the previous section, it is reasonable to assert the robustness of the fluvial flood hazard maps. Naturally, future improvements in the quality and quantity of the available input datasets along with advancements in digital elevation resolution, are expected to help reducing the uncertainty further.




Th fluvial flood hazard maps for the current climate conditions have been computed over the entire Central Asia area and specifically for the five countries of Kazakhstan, Kyrgyz Republic, Uzbekistan, Tajikistan, and Turkmenistan for the selected eight return periods of 5, 10, 20, 50, 100, 200, 500, 1000 years. Two sets of fluvial flood hazard for current climate conditions were computed, namely for the undefended and defended scenarios. Figure 12 and Figure 13 show some of the resulting hazard maps for a return period of 100 years.

The comprehensive collection of the computed results of the flood hazard model can be found here: https://datacatalog.worldbank.org/search?q=SFRARR&sort=&start=0.

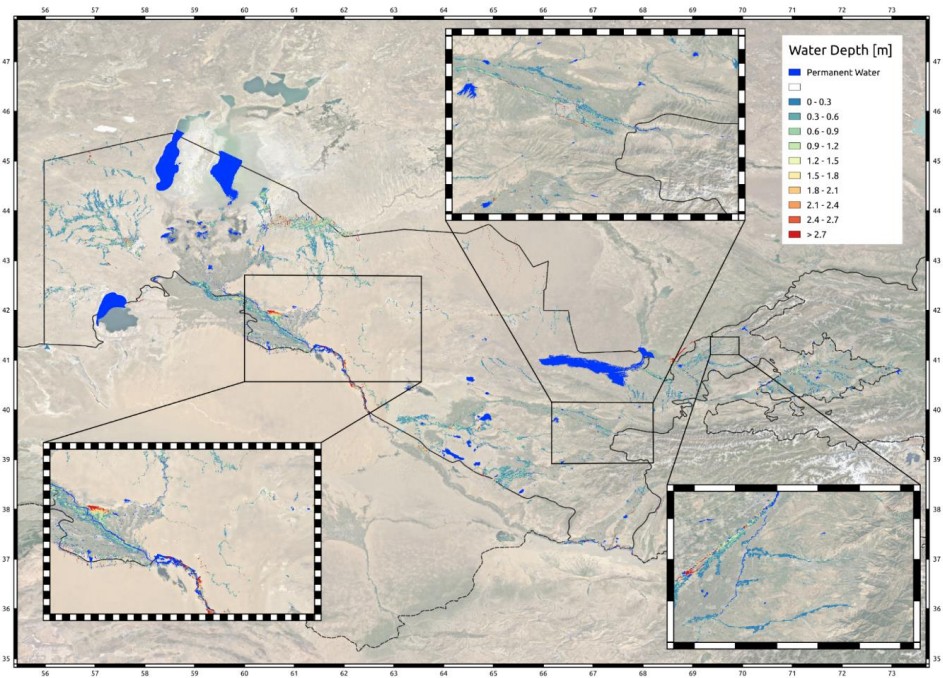

**Figure 12. Fluvial flood hazard map for Uzbekistan, 100-year return period, undefended scenario**
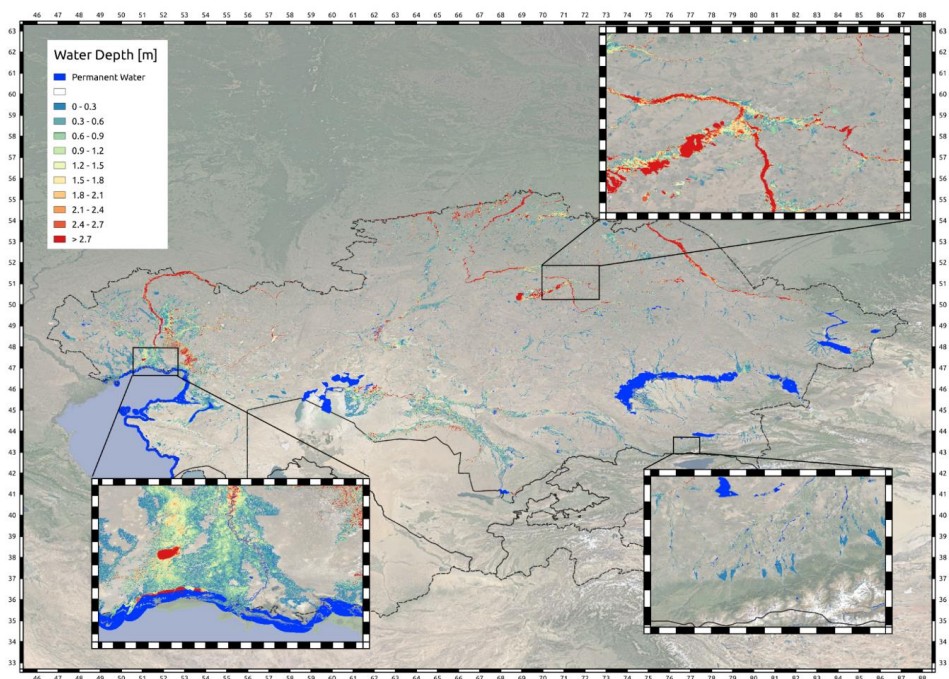

**Figure 13. Fluvial flood hazard map for Kazakhstan, 100-year return period, undefended scenario**

**5.2.2 Hazard curves for selected target cities**

Figure 14 shows the derived fluvial flood hazard curves for undefended conditions for five selected locations within the urban areas of the main flood-prone cities of the target countries: Turkmenabat, Tashkent, Dushanbe, Nur Sultan, Bishkek.


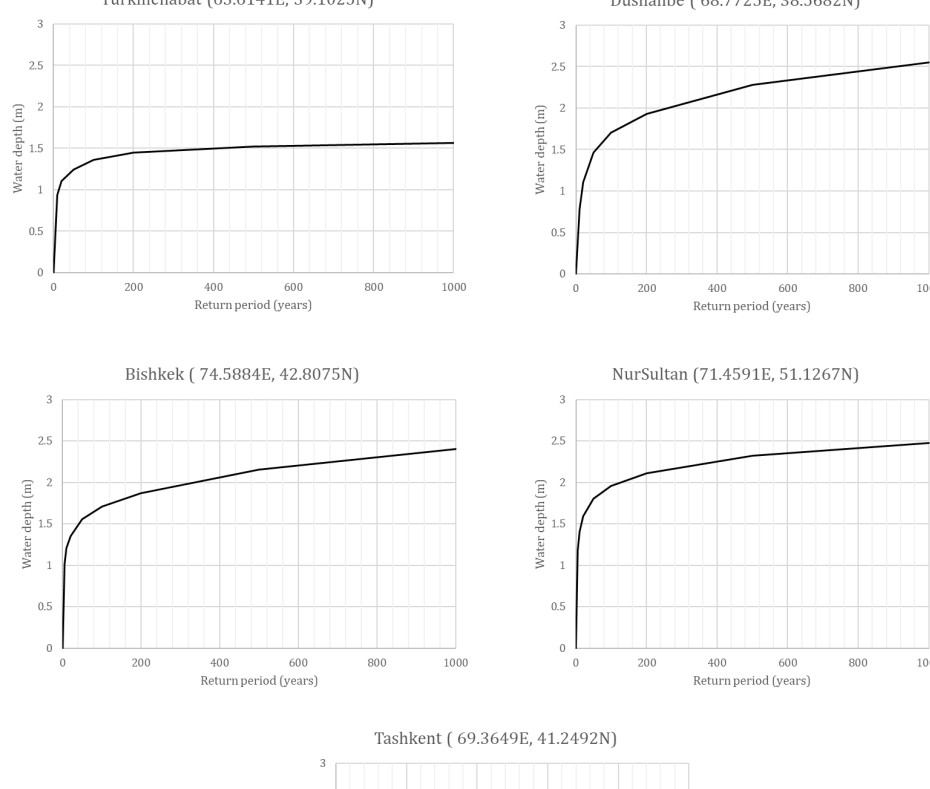

**Figure 14. Undefended flood hazard curves computed at five selected target sites**


### 5.3 Flood risk

The flood risk results were computed following the risk assessment methodology previously described. Although this article discusses only the flood-induced losses, as alluded to before this effort was part of a multi-hazard risk assessment study that included also assessing risk for earthquakes (for the earthquake risk assessment please refer to Salgado et al. (2023). To compare the risk across different perils, it was necessary to use a peril-agnostic assessment methodology such as that adopted in CAPRA, which uses a common representation of the disaster risk assessment components, i.e., hazard, exposure, and vulnerability. Since the simultaneous occurrence of earthquakes and floods (at least of those causing the large losses of interest to stakeholders) is highly unlikely, the losses caused by the two types of events have been assumed to be independent.



### 5.3.1 Risk metrics

The results of the flood risk assessment are presented in terms of a loss exceedance probability curve (EP curve) and by the year loss tables (YLTs), disaggregated at administration units 1 (ADM1, which is equal to Oblast level) and administration unit 0 (ADM0, which is equal to country level). Furthermore, return period loss estimates and Average Annual Loss (AAL) at ADM1 and ADM0 levels, and for the whole region are provided in tabular format for the same eight return periods ranging from 5 to 1000 years reported earlier. In addition, for preparedness and mitigation plans it is important to estimate the possible losses (economic and human) that scenario events may cause to the current exposure. To be consistent with the hazard and representative of scenarios that may actually strike the five capitals of these countries, these as-if scenarios were selected based on the results of the earthquake hazard disaggregation and the flood hazard analyses. The loss results have been derived both in terms of expected values and their confidence intervals. Finally, exposure levels to various hazard intensity thresholds have been assessed for population, key industrial sites, critical infrastructure, and supply infrastructure. These results are available for the current conditions (year 2020) and future (2080) scenarios considering three different projections to year 2080: the three Shared Socio-Economic Pathways SSP1, SSP4 and SSP5 considered in the exposure model development. The future exposure only considers the residential sector. Losses have been calculated for physical risk (monetary) and human risk (fatalities).

### 5.3.2 Probabilistic flood risk results

Table 4 shows the fluvial flood risk results (undefended case) at national and regional levels in absolute and relative (to the total replacement cost of the exposure dataset) terms for the current exposure scenario. The highest absolute fluvial risk is found to be in Kazakhstan and Uzbekistan. However, when assessed in relative terms, Kazakhstan, Tajikistan and Turkmenistan have similar risk values with an AAL above 2‰. The same results for the defended case shown in Table 5 highlight a large risk reduction especially for Kazakhstan, Turkmenistan and Uzbekistan. Table 6 shows the fluvial flood risk results at country and regional levels (undefended case) for one of the three different projected scenarios with consideration of the effect of climate change. The aforementioned tables use the country ISO3 codes: KGZ [Kyrgyz Republic], KAZ [Kazakhstan], TJK [Tajikistan], TKM [Turkmenistan], and UZB [Uzbekistan].

The most substantial variations among the examined scenarios stem from whether flood defenses are factored in or not, revealing more consistent disparities. In contrast, the influence of climate change, while noteworthy, exhibits greater variability depending on the specific geographical context. The exposure dataset used in the flood risk assessment for the 2080 projection only includes the residential sector, although in terms of absolute losses, the differences between the current scenario (that includes all lines of business) and the 2080 scenario (only residential assets) are not as large.

As a general comment on the estimate of flood losses, it appears that the exceedance probability curves tend to saturate relatively quickly (i.e., the slope of the curve is decreasing sharply with increasing frequency). There are three factors contributing to the quick saturation of the EP curves, one related to the hazard, another to the vulnerability, and a third one to the reproduction of flood defenses. First, flood depth hazard curves (i.e., relationships between flood depth and frequency) in this region often have a rather "flat" shape (i.e., the increase in flood depth with frequency is gradually smaller with high frequency); this phenomenon is typical of frequently inundated flat areas where floods are rather common, but the difference between water depths for small intensity and high intensity events is not that large, due to the fact that large alluvial floodplains provide plenty of space for water propagation. Second, the flood vulnerability curves developed for this project typically saturate at 30%-40%-50% of the total exposed value, mainly because they represent asset classes that bundle together buildings with different number of stories. This means that losses after a certain water depth increase very slowly, therefore causing a saturation of the loss vs frequency curve. This is typical of losses calculated for assets spread out in large-scale regions, some of which are exposed to high flood risk and others are relatively safe. Finally, another important issue is the inclusion of flood defenses in the model: a reliable representation of the flood defenses in the model would necessarily lower the high-frequency losses. However, very little data were available to precisely reproduce the flood defenses in the region, and therefore the results of the model are considered to be conservative, especially in the high-frequency part of the EP curve. Because of the characteristics of the region (many large




fluvial plains with little population) and the model (large-scale aggregation and unavailability of data regarding flood defenses), we believe that the quick saturation of the flood loss curves is justified.

**Table 4. Losses for different return periods (first 9 lines) and AAL (last line) for fluvial flood risk undefended scenario at Regional and Country level. Grey columns show the absolute value in USD and red columns show the relative losses in per mille. The results were computed for the 2020 total exposure.**

| Tr (years) | Absolute values ($Million USD) | | | | | Relative values to the total replacement cost (per mille) | | | | | |
|---|---|---|---|---|---|---|---|---|---|---|---|
| | Regional | KGZ | KAZ | TJK | TKM | UZB | Regional | KGZ | KAZ | TJK | TKM | UZB |
| 5 | $2,664.3 | $130.7 | $1,522.0 | $242.0 | $203.2 | $867.2 | 1.60 | 2.31 | 2.72 | 3.25 | 3.69 | 0.94 |
| 10 | $2,988.6 | $156.6 | $1,755.9 | $292.7 | $262.9 | $1,037.9 | 1.79 | 2.77 | 3.14 | 3.93 | 4.77 | 1.12 |
| 25 | $3,360.0 | $185.6 | $2,021.3 | $349.7 | $342.5 | $1,240.2 | 2.01 | 3.28 | 3.62 | 4.70 | 6.22 | 1.34 |
| 50 | $3,595.7 | $205.3 | $2,197.2 | $381.8 | $393.0 | $1,380.9 | 2.15 | 3.63 | 3.93 | 5.13 | 7.13 | 1.49 |
| 100 | $3,797.4 | $224.0 | $2,361.5 | $409.5 | $437.0 | $1,527.5 | 2.27 | 3.96 | 4.22 | 5.50 | 7.93 | 1.65 |
| 250 | $4,024.7 | $241.4 | $2,605.0 | $449.1 | $502.8 | $1,760.0 | 2.41 | 4.26 | 4.66 | 6.03 | 9.13 | 1.90 |
| 475 | $4,178.7 | $249.7 | $2,830.8 | $480.4 | $557.3 | $1,897.5 | 2.50 | 4.41 | 5.06 | 6.45 | 10.12 | 2.05 |
| 500 | $4,190.7 | $250.3 | $2,845.7 | $483.0 | $561.2 | $1,908.5 | 2.51 | 4.42 | 5.09 | 6.49 | 10.19 | 2.06 |
| 1000 | $4,354.0 | $257.6 | $3,004.5 | $515.6 | $610.3 | $2,031.3 | 2.61 | 4.55 | 5.37 | 6.93 | 11.08 | 2.20 |
| AAL | $2,190.9 | $95.1 | $1,165.6 | $177.0 | $123.0 | $630.2 | 1.31 | 1.68 | 2.09 | 2.38 | 2.23 | 0.68 |

The graphical representation of the results of Table 4 is shown in Figure 15 (return period losses: absolute values to the left, relative values to the total replacement cost in the center; AAL absolute values to the right).

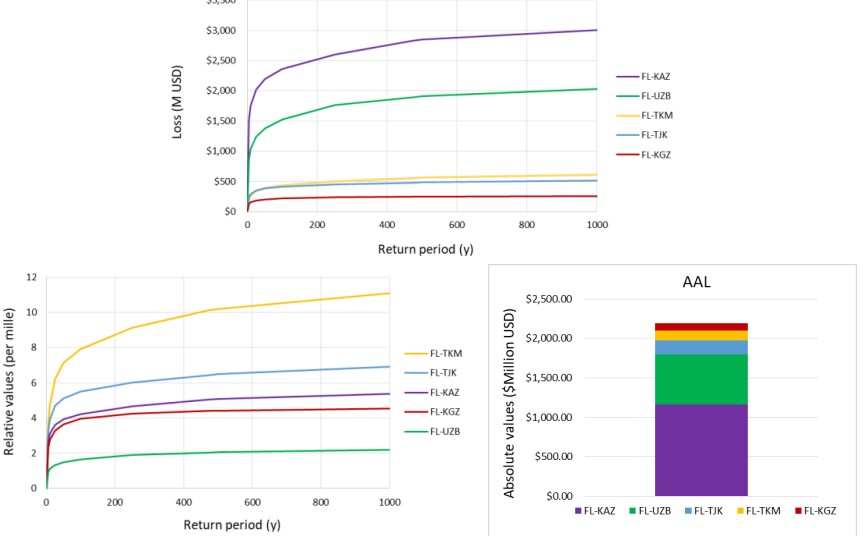

**Figure 15. Return period losses and AAL (absolute and relative values to the total replacement cost) at Country level for fluvial flood risk, undefended case, 2020 exposure**


**Table 5. Losses for different return periods (first 9 lines) and AAL (last line) for fluvial flood risk defended scenario at Regional and Country level. Grey columns show the absolute value in USD and red columns show the relative losses in per mille. The results were computed for the 2020 total exposure.**

| Tr (years) | Absolute values ($Million USD) | | | | | Relative values to the total replacement cost (per mille) | | | | | |
|---|---|---|---|---|---|---|---|---|---|---|---|
| | Regional | KGZ | KAZ | TJK | TKM | UZB | Regional | KGZ | KAZ | TJK | TKM | UZB |
| 5 | $1,876.9 | $124.8 | $976.6 | $237.6 | $150.5 | $612.3 | 1.12 | 2.20 | 1.75 | 3.19 | 2.73 | 0.66 |




| Tr | | | | | | | | | | | |
|---|---|---|---|---|---|---|---|---|---|---|---|
| 10 | $2,170.6 | $150.0 | $1,200.6 | $288.3 | $207.2 | $786.7 | 1.30 | 2.65 | 2.15 | 3.87 | 3.76 | 0.85 |
| 25 | $2,481.9 | $179.2 | $1,474.3 | $345.7 | $282.7 | $993.5 | 1.49 | 3.16 | 2.64 | 4.64 | 5.13 | 1.07 |
| 50 | $2,677.5 | $197.7 | $1,655.5 | $378.0 | $336.2 | $1,126.7 | 1.60 | 3.49 | 2.96 | 5.08 | 6.10 | 1.22 |
| 100 | $2,871.1 | $215.7 | $1,807.9 | $404.2 | $380.3 | $1,253.2 | 1.72 | 3.81 | 3.23 | 5.43 | 6.90 | 1.35 |
| 250 | $3,145.9 | $232.4 | $2,030.0 | $445.4 | $443.7 | $1,435.5 | 1.88 | 4.11 | 3.63 | 5.98 | 8.06 | 1.55 |
| 475 | $3,322.4 | $240.7 | $2,207.4 | $479.2 | $483.7 | $1,542.6 | 1.99 | 4.25 | 3.95 | 6.44 | 8.78 | 1.67 |
| 500 | $3,335.5 | $241.2 | $2,222.1 | $482.0 | $486.3 | $1,550.7 | 2.00 | 4.26 | 3.98 | 6.47 | 8.83 | 1.68 |
| 1000 | $3,519.0 | $248.6 | $2,387.8 | $513.3 | $522.2 | $1,657.8 | 2.11 | 4.39 | 4.27 | 6.90 | 9.48 | 1.79 |
| **AAL** | $1,513.7 | $91.0 | $726.6 | $173.7 | $89.40 | $432.96 | 0.91 | 1.61 | 1.30 | 2.33 | 1.62 | 0.47 |


The graphical representation of the results of Table 5 is shown in Figure 16 (return period losses: absolute values to the left, relative values to the total replacement cost in the center; AAL absolute values to the right).

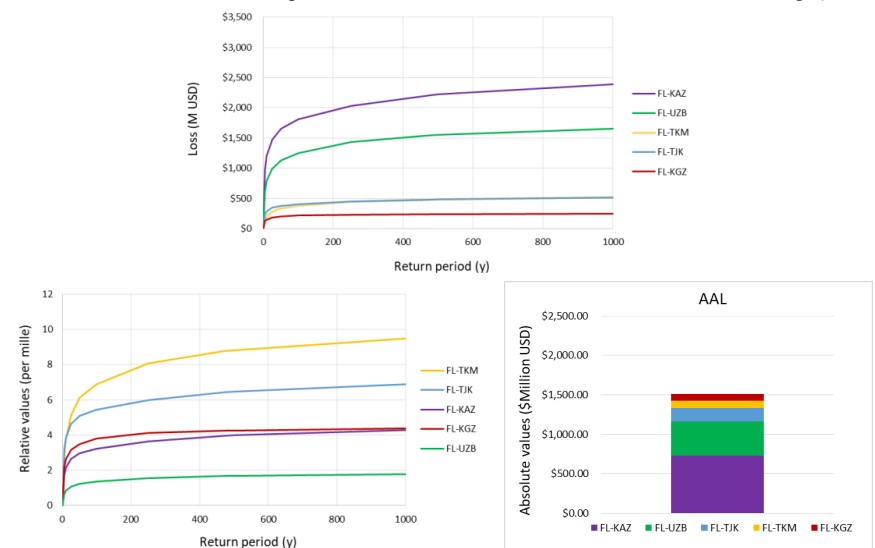

**Figure 16. Return period losses and AAL (absolute and relative values to the total replacement cost) at Country level for fluvial flood risk, defended case, 2020 exposure**

**Table 6. Losses for different return periods (first 9 lines) and AAL (last line) for fluvial flood risk undefended scenario at Regional and Country level. Grey columns show the absolute value in USD and red columns show the relative losses in per mille. 2080 SSP1 exposure (residential only).**

| Tr (years) | Absolute values ($Million USD) | | | | | Relative values to the total replacement cost (per mille) | | | | | |
|---|---|---|---|---|---|---|---|---|---|---|---|
| | Regional | KGZ | KAZ | TJK | TKM | UZB | Regional | KGZ | KAZ | TJK | TKM | UZB |
| 5 | $1,800.1 | $122.8 | $1,168.7 | $44.4 | $36.4 | $564.7 | 1.59 | 3.81 | 3.47 | 0.80 | 1.83 | 0.82 |
| 10 | $2,028.4 | $146.1 | $1,354.8 | $53.9 | $51.9 | $698.5 | 1.79 | 4.54 | 4.03 | 0.97 | 2.61 | 1.02 |
| 25 | $2,315.8 | $169.2 | $1,573.3 | $63.2 | $72.3 | $842.0 | 2.05 | 5.25 | 4.67 | 1.13 | 3.63 | 1.23 |
| 50 | $2,512.8 | $183.4 | $1,730.6 | $69.3 | $90.5 | $942.9 | 2.22 | 5.69 | 5.14 | 1.24 | 4.55 | 1.37 |
| 100 | $2,705.4 | $197.1 | $1,884.7 | $74.9 | $105.5 | $1,055.4 | 2.39 | 6.12 | 5.60 | 1.34 | 5.30 | 1.54 |
| 250 | $2,928.1 | $211.7 | $2,121.0 | $81.4 | $123.6 | $1,196.8 | 2.59 | 6.57 | 6.30 | 1.46 | 6.21 | 1.74 |
| 475 | $3,054.5 | $221.6 | $2,267.7 | $85.1 | $134.9 | $1,267.2 | 2.70 | 6.88 | 6.74 | 1.53 | 6.78 | 1.85 |
| 500 | $3,062.7 | $222.3 | $2,276.5 | $85.3 | $135.9 | $1,271.9 | 2.71 | 6.90 | 6.76 | 1.53 | 6.82 | 1.85 |
| 1000 | $3,174.3 | $232.2 | $2,390.9 | $87.7 | $147.7 | $1,335.3 | 2.81 | 7.21 | 7.10 | 1.57 | 7.42 | 1.94 |
| **AAL** | $1,432.2 | $78.2 | $884.8 | $29.5 | $23.8 | $416.2 | 1.27 | 2.43 | 2.63 | 0.53 | 1.20 | 0.61 |


Some of the relative losses computed for the current (2020) and future (2080) scenarios are plotted in Figure 17, Figure 18, and Figure 19. The risk reduction immediately apparent when comparing the results of Figure 17 and Figure 18 is due to the inclusion of flood defenses.

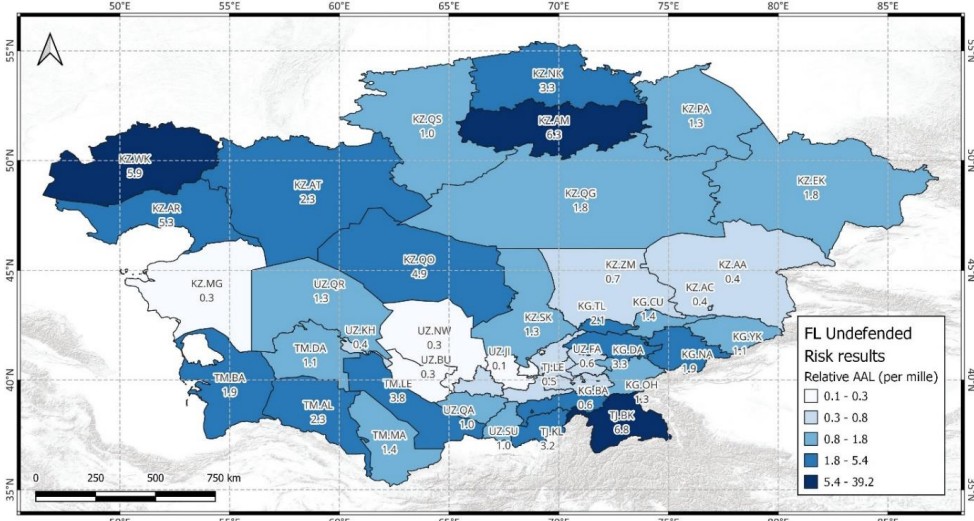

**Figure 17. Current (2020) exposure and undefended scenario: relative losses**

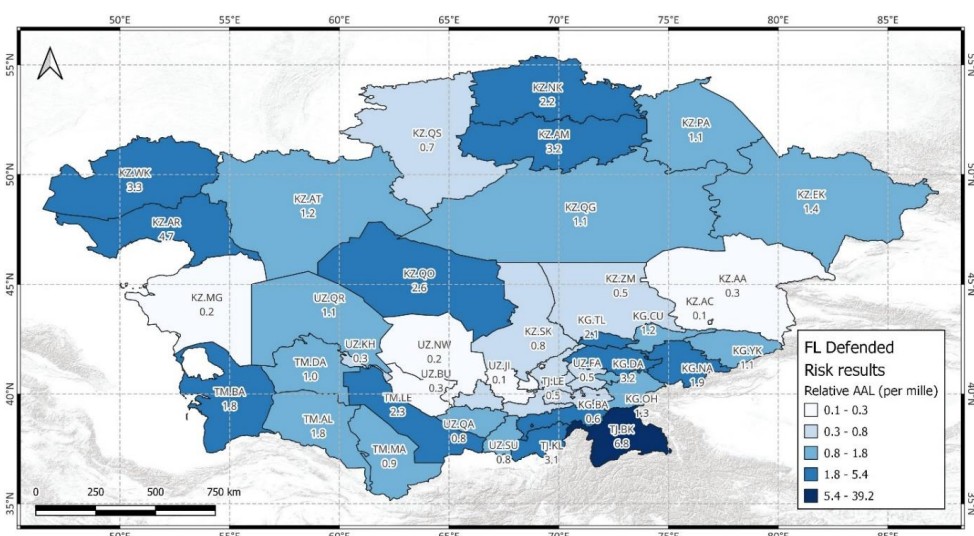


**Figure 18. Current (2020) exposure and defended scenario: relative losses**

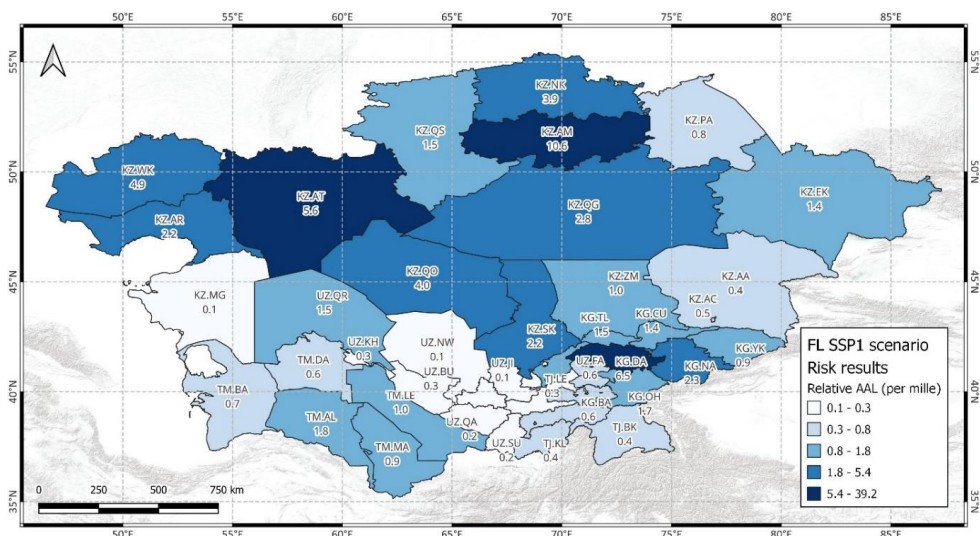

**Figure 19. Future (2080) exposure and Climate Change – SSP1 scenario: relative losses**

For the undefended scenario, the largest relative AALs are found in Kazakhstan and Tajikistan, with values above 6‰. In the five considered countries, the largest relative AALs by sector are found for the transport and agricultural sector (the two types of crops included in this assessment: cotton and wheat). For the case of cotton crops, the largest relative AALs are found in Kazakhstan, Turkmenistan, and Tajikistan with values above 6‰. Regarding flood risk fatalities, the highest risk is found, as expected, for the undefended case. The largest values

are found for the Akmolinskaya Oblast in Kazakhstan and Khatlon Province in Tajikistan. On average, at Oblast level, there is a decrease of 20% of flood fatalities' risk in the defended case. Regarding future scenarios, and considering climate change, there is a variable trend at Oblast level for the flood fatality risk, although consistent among the considered SSPs. In general, risk values in future scenarios are increased by a factor between 1.5 and 2.0, such as in the following Oblasts: Sirdarya (Uzbekistan), Ysyk-Kol, and Jalal-Abad in the Kyrgyz Republic

and Turkistan and Karagandiskaya in Kazakhstan. However, there are extreme cases, such as the Mangistauskaya Oblast in Kazakhstan, where the risk increases by seven times. Conversely, , there are Oblasts, such as Lebap (Turkmenistan), Khatlon Province (Tajikistan), Samarkand (Uzbekistan) and Batken (Kyrgyz Republic), for which decreases between 80 and 90% are observed for all SSPs.

   As expected, flood risk is lower for the defended case although caution should be used when interpreting these

results due to the assumptions about flood defenses' location and height discussed earlier. That being said, a comparison between the two cases at Oblast level can be made and some discussion is provided next. Overall, the Oblast with the largest flood AAL is the Badakhshan Autonomous Mountainous Region in Tajikistan. The largest relative difference caused by modelling or not the flood defenses is found in Batken Oblast (Kyrgyz Republic), although for the undefended case the flood risk AAL was relatively low (0.4‰). A major flood risk reduction due

to the inclusion of the defenses is observed in Ysyk-Kol Oblast, with a decrease of around 40%, which is a significant decrease considering the large flood risk AAL for the undefended case.

   As previously stated, these results are part of a multi-hazard risk assessment study including both earthquake and flood risk. Combined results (earthquake and undefended flood), show that there are Oblasts such as JalalAbad (in the Kyrgyz Republic), and Khatlon Province and the Cities and Districts of the Republican Subordination (in

Tajikistan) where the risk caused by these two hazards is high. On the other hand, there are Oblasts where only the risk caused by one of these hazards is relevant whereas the other is very low, as in the case of Zapadno-kazachstanskaya (in Kazakhstan) where the undefended flood relative AAL is almost 6‰ and earthquake relative AAL is approximately 0.1‰, or the case of Almatinskaya Oblast (in Kazakhstan) where earthquake relative AAL is above 2‰ but undefended flood relative AAL is lower than 0.5‰. Similar results can be observed for the Sughd

Province (Takikistan) and Namangan (Uzbekistan) with large and low relative AALs for earthquakes and floods, respectively, or for Karakalpakstan (Uzbekistan), Lebap (Turkmenistan) and Akmolinskaya (Kazakhstan), with


large and low relative AALs for floods and earthquakes, respectively (for further details on the earthquake risk assessment please refer to Salgado et al. (2023).

## 6 Discussion

This study presents the first high-resolution, regional-scale, transboundary fully probabilistic risk assessment for the area, providing decision-making aids and disaster risk management resources. Notably, the involvement of local stakeholders and unprecedented access to local data enhance its significance.
Hereafter, we delineate a series of strengths and limitations inherent to this risk assessment.

### 6.1 Strengths

- A main strength of this risk assessment is that a peril-agnostic methodology was used, facilitating the comparison of results across country, sector, and hazards (earthquake and flood). This was achieved by using the same representation for all the key risk components and by computing the same risk metrics using the same probabilistic approach. The earthquake losses are discussed in detail in Salgado et al. (2023).

- This is the first study in the region that disaggregates flood risk results into subnational level (Oblast), national level (country), and regional level (five countries), providing a complete disaster risk estimation and results compatible with the overall objectives of the project.

- The regional approach adopted for carrying out this risk assessment used consistent assumptions, modelling approaches and treatment of uncertainties. This is key considering that the final objective of
this study is the regional calculation of losses caused by floods of different types (pluvial not shown here) and different kind of events (earthquakes).

- This is the first project in the region that considers a complete exposure dataset for the estimation of flood risk. Besides buildings (considered in previous studies), other relevant types of assets, such as the transportation infrastructure (roads and bridges) and key crops (cotton and wheat) have been included
too.

- Given that the software utilized to estimate the physical and human losses has a friendly graphic user interface and some GIS capabilities, the obtained flood risk results are expected to facilitate the capacity building process in disaster risk assessment in Central Asia.

- The risk results obtained in this study provide losses for floods at subnational level with a reasonable
level of accuracy. This has been achieved using a good amount of local data for hazard modeling and risk validation and adopting a high-resolution approach to the modeling of the hazard and exposure components.

- By having developed an exposure dataset with different lines of business, all the loss results can be disaggregated by categories. This information is valuable to different stakeholders at subnational to
regional levels.

- The level of detail paid to most components of the flood risk model is higher than that adopted in previous studies carried out in the region. The refined approach has been complemented by the inclusion of additional lines of business in the exposure datasets, an addition that enabled the derivation of a more comprehensive picture of the flood risk in the region.

### 6.2 Limitations

- The hazard model is not supported by the level of detail and accurate data that are often available when developing national and sub-national scale models in other regions. However, until those more detailed analyses are performed and made available to the public (https://datacatalog.worldbank.org/search?q=SFRARR&sort=&start=0), these risk estimates can
certainly be used as first-order quantification of risks. These risk estimates are certainly suitable both to support raising awareness on this topic, and to guide the development of more refined analyses with the same probabilistic framework adopted here.

- Catastrophe risk models always have an associated level of uncertainty even when developed for the current hazard and exposure characteristics. In this project, a projection of exposure was performed to



year 2080 (for the residential sector only) for different Shared Socio-Economic Pathways (SSPs) for one climate change scenario. These results are intended to be indicative and useful for comparison purposes only. The relative results should be preferred over the absolute losses.

- The risk estimates should not be used as the only support for planning and designing specific risk management infrastructure. These applications should be informed by flood risk studies for specific areas
that utilize a more comprehensive set of input data such as those confidential and highly classified datasets available to the central governments.

## 7 Conclusions

This article presented the methodological framework utilized for developing a fully probabilistic flood risk assessment for Kazakhstan, Kyrgyz Republic, Tajikistan, Turkmenistan and Uzbekistan in Central Asia and the
obtained risk estimates. The results are expressed in terms of EP curves, AAL and specific return period losses, which are the metrics commonly used to shape different disaster risk management strategies. The risk assessment study includes several variants of the hazard (current and future including climate change conditions) and exposure components (current all lines of business and future, residential line only, for different Shared Socio-Economic Pathways). The results of the risk assessment are of general use but were intended primarily to inform World
Bank's engagement in supporting regional and national disaster risk financing and insurance applications, including traditional and parametric solutions for the structuring of a regional risk mitigation program. These risk estimates can be used by the World Bank to initiate a policy dialogue with the governments of Kazakhstan, Kyrgyz Republic, Tajikistan, Turkmenistan and Uzbekistan.

## Acknowledgements

This study has been carried out within the framework of the World Bank-funded project "Strengthening Financial Resilience and Accelerating Risk Reduction in Central Asia" (SFRARR). The authors would like to thank all the local experts involved in the project tasks and the participants to the workshops organised within the SFRARR project for their contribution to the implementation of the flood risk model presented in this paper. The constant support of Dr. Sergey Tyagunov in all project tasks was greatly appreciated.

## Competing interests

The contact author has declared that none of the authors has any competing interests.

## Data availability

The technical reports produced during the SFRARR project, containing detailed explanations of the work carried out during the fully probabilistic flood risk assessment, are available upon request to the corresponding author.

## Author Contributions

GC, GB, SD: conceptualization, validation, risk assessment methodology, formal analysis, writing review and editing. PC: conceptualization, data collection, validation, writing and editing. PB: risk assessment methodology, validation, calibration, writing review and editing; MM, EF, MO: conceptualization, risk assessment methodology. CA, BH, OG: software, risk assessment. ZR, KA, SM, VI: data collection, calibration, validation. VB: data
collection, calibration, validation, writing review.



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
