# Peer review of "Large-scale flood risk assessment in data scarce areas: an application to Central Asia"

_Natural Hazards and Earth System Sciences, 2023_

## Author Response (AR1)

**Comments by Reviewer n. 1 and Replies by Authors**

The paper "Large-scale flood risk assessment in data scarce areas: an application to Central Asia" presents a comprehensive analysis in a region that is rarely given attention in natural hazards research. The authors identified the best available global datasets, combined them with local sources and connected the whole through a long modelling chain. The study is mostly sound methodologically and is an important contribution. However, the paper itself is not well structured and quite difficult to navigate. Therefore, a lot of my comments pertain to structuring of the paper as well as too much of some details, the lack of certain other details, and terminology used. Below, I describe the main issues found in the sections, then I discuss some overarching issues, and end with some minor comments.

R: Thank you for your thorough review and valuable feedback on our paper, "Large-scale flood risk assessment in data scarce areas: an application to Central Asia." We appreciate your recognition of the importance of our study in addressing natural hazards research in a region often overlooked. We acknowledge the need for improved organization and clarity in the paper's structure, as well as addressing concerns regarding the level of detail and terminology used.

We will take your suggestions into careful consideration and reorganize the paper to enhance its navigability and coherence. Your insights will undoubtedly contribute to refining the presentation of our research findings. We are committed to addressing the overarching issues you've raised while ensuring that the paper remains methodologically robust and impactful. Once again, we sincerely appreciate your thoughtful review, and we look forward to incorporating your suggestions to improve the overall quality of our manuscript.

**Abstract**

The last paragraph is rather out of place and should be in introduction or conclusions. Please replace this paragraph with some of the results of the assessment, including your projected climate change impacts.

R: We acknowledge your point regarding the placement of the last paragraph and agree that it would be more suitable either in the introduction or conclusions section. In our revised manuscript, we will relocate this paragraph accordingly. Additionally, we will incorporate here some of the results from the assessment, including our projections of climate change impacts, as suggested.

**Section 3.1**

The datasets are well-known and widely used, so the section can be reduced to single paragraph that refers to details in the table. References and spatial resolution should be mentioned in the table. Fig. 2 is not needed, as you have 19 figures and it adds nothing to the analysis. Climate projections should be mentioned in the table, and the paragraph on this moved to section 4.1.

R: We appreciate your feedback on the dataset section and acknowledge the widespread use of the datasets we employed. Therefore, we agree to condense this section into a single paragraph that directs readers to the pertinent details provided in the table, including references and spatial resolution.

In line with this, we will incorporate climate projections information into the table for better clarity and accessibility. Additionally, we concur with your suggestion to remove Fig. 2, as it does not significantly contribute to the analysis given the ample number of figures already present.

"We used observed data from the KNMI Climate Explorer to assess and correct the ERA5-Land extreme precipitation estimates due to the discrepancy between point station data and grid averaged data". The authors do not elaborate or cite any literature here, despite potentially significant influence on the results. Bias-adjusting climate data is a major undertaking and don't see how the authors did it having only observational point data.

R: Regrettably, the bias correction of ERA5-Land precipitation using raingauge data is not relevant to the present paper and should not have been placed in the draft originally submitted. The original study carried out fluvial and pluvial flood risk assessments, but the present paper only shows fluvial risk assessment. This bias correction mentioned here was not carried out for the fluvial risk assessment, only for the pluvial risk assessment, and therefore is not relevant here. We apologise for the confusion. We will remove all references to pluvial flood and ERA5-Land bias correction from the paper.

**Section 3.2**

The text on how the data were obtained (or not obtained) could be shortened, with non-essential information moved to discussion. It would be best to merge Fig. 3 and 4, and use a different colour for population, as the bright blue will make everybody immediately assume it shows the flood hazard map.

R: We understand your point about shortening the text on how the data were obtained and moving non-essential information to the discussion section for better flow and clarity. We will revise the text accordingly in our manuscript.

We will change the figure colour and agree that the two figures should be merged.

Further, the paragraph from L203 is really difficult to understand in terms of what data were collected (extent, resolution, timeliness) and what is its use in the model.

R: We will streamline paragraph from L203 to clearly elucidate the data collection process, including details on extent, resolution, timeliness, and its relevance to the model.

**Section 3.3**

I think that this section should be limited to the hydrological models. The other paragraphs are very confusing as to what are your exposure data and how the vulnerability models look like as some local modifications are mentioned, but not described. The authors should rather continue with section 4.1 after the hydrological model description, and then create a new section that collects together information on your exposure data and vulnerability models that is now spread across the paper, and then continue with section 4.2. In this way, a clear structure will appear: flood hazard modelling described comprehensively in section 3, and the transition to flood risk in section 4.

R: All the details related to the developed exposure model and the assets at risk included in the model can be found in two companion papers (Scaini et al., 2023 – for population and residential buildings; Scaini et al., 2024 – for non-residential buildings, transportation infrastructure and cropland).

**Section 4.1**

The section first summarizes the methods in a figure, then in a list and then a full description. I suggest to remove the list and divide the rest of the section into subsections corresponding to the 5 steps. As noted earlier, the climate change aspect should have its own subsection with a full description.

R: Thank you for your suggestion. We will revise Section 4.1 to remove the list and reorganize it into subsections corresponding to the 5 steps, including a dedicated subsection for the climate change aspect with a detailed description.

Here, there a crucial aspect of how the authors created the climate projection dataset. It is a very practical "solution" to modify ERA5 as if it was being bias-adjusted. However, only one reference is cited, which neither covers climate changes nor temperature. I haven't

seen this approach used for climate change projections, which normally use hindcasts for both historical and future periods, for modelling consistency. The authors should elaborate here and provide more explanation why this approach is used and whether it was applied in literature before.

R: Bias-correcting climate projections before using them in hydrological modelling is standard practice and should always be carried out to avoid propagating the climate model biases into the hydrological model results (Shrestha et al., 2017; Teutschbein and Seibert., 2012). The methodology we used here belongs to the "delta change" family cited by Teutschbein and Seibert (2012), The literature on this methodology and its implications on hydrological model outputs is very extensive and well documented, here we cite only a few examples (Räty et al., 2014; Mudbhatkal and Mahesha, 2017; Räty et al., 2018; Fang et al., 2015). It is simpler than other techniques, since it does not require to bias-correct the baseline climatology (which is still the observed climatology), although it has the disadvantage that some properties of the variable to be corrected still remain unadjusted (for example, if the precipitation from a certain climate projection is simply multiplied by a factor in order to reproduce the annual average of the reference dataset, the distribution of the original reference dataset will be maintained and only the mean values will be corrected – this is also called "constant scaling"). However, the approach used in this paper, which adjust the whole distribution of precipitation and temperature, not only the mean or the standard deviation, limits this disadvantage. Räty et al. (2014), among others, have discussed the advantage and disadvantages of such technique, which blends the simplicity of the delta factor methodologies with the robustness of the quantile mapping methodologies.

Flood protection: the authors scale the protection using correlation between FLOPROS data and population density. However, FLOPROS itself doesn't contain any actual data for Central Asia: it was created using correlation with GDP per capita. Then, the authors combine it with some local data. Due to the influence of flood protection on results, the map of the protection assumptions should be presented, or the data made available online. The way it is now, I can't really assess this aspect of the authors' work.

R: Thank you for your comment. We appreciate the opportunity to clarify our methodology. Contrary to the interpretation that we integrated FLOPROS standards with local data, our approach involved developing a strategy to derive the hydraulic protection level based on the correlation between protection level and population density along the river. Initially, we identified urban agglomerates and determined their maximum population density through data from HBASE (Wang et al., 2017b) and WorldPop (Tatem, 2017). Then, we associated river segments with each urban area, assuming uniform urban protection levels within variable distances from the agglomerate, determined by accumulated area size. Finally, we employed a linear scaling method using FLOPROS area protection standards to refine our estimates. While we did integrate localized qualitative data for validation purposes, it did not directly influence our methodology but rather served to validate our approach.

We will clarify this strategy in the revised manuscript.

**Section 4.2**

I'm missing here information on what are the asset types covered, and how were the damage functions derived. How do the authors know that they are applicable to Central Asia?

R: The assets covered are buildings, infrastructure (roads and airport) and crops. More information can be found in two companion papers (Scaini et al., 2023; Scaini et al., 2024).

[revised manuscript text omitted]

Paragraph L407-416 is generally duplicated from the previous, though with some minor changes. Please verify which version is correct and remove the other one.

R: Thank you for pointing out the duplication in paragraph L407-416. We will carefully review both versions and ensure that only the correct one remains in the manuscript.

**Section 5.1.1**

Description of calibration in L458-471 should be in the methodology.

Additionally, the authors should describe what parameters were subject to calibration (or provide suitable reference) as well as what period was calibrated.

Then, description in L471-483 should be in the discussion. Then, though the authors show example figures, no statistical analysis is presented.

The figures should rather go to a supplement, and replaced with tables or graphs showing summary performance of the calibration & validation indicating correlation, bias and/or metrics.

R: Thank you for your feedback. We will move the description of calibration from L458-471 to the methodology section and provide clarification on the calibrated parameters and calibration period: for the hydrological simulation, we assumed two soil layers (superficial and sub-superficial) and the parameters that were calibrated included horizontal conductivity and depth for each of the two layers, vertical conductivity, potential evapotranspiration and snowmelt rate. Furthermore, we will provide details on the calibration period, which varied among historical stations, with record lengths ranging from 15 to 37 years.

Additionally, we will move the discussion from L471-483 to the discussion section.

Regarding statistical analysis, we acknowledge the need for summary performance metrics. We will transfer the figures to a supplementary section and replace them in the main text with the following table displaying summary performance metrics of the calibration process: the table shows the correlation and the percent bias on the streamflow that were adjusted with the correction procedure described in the paper for 30 stations distributed across the region.

| STATION | CORRELATION | BIAS_PERCENT_CORR |
|---------|-------------|-------------------|
| KAZ_158 | 0.12182397 | -15.78521 |
| KAZ_160 | 0.3399278 | 19.1863648 |
| KAZ_161 | 0.61834251 | 3.33038515 |
| KAZ_165 | 0.72464942 | 8.66085851 |
| KAZ_166 | 0.71786685 | 1.9393047 |
| KAZ_172 | 0.29560686 | 13.6315298 |
| KAZ_232 | 0.84705051 | -10.582237 |
| KAZ_233 | 0.39452119 | -55.398565 |
| KAZ_234 | 0.5618009 | -6.9184438 |
| KAZ_235 | 0.32640419 | -17.83126 |
| KAZ_238 | 0.72432363 | -7.3196522 |
| KAZ_227 | 0.42705552 | 8.32165952 |
| KAZ_228 | 0.73364143 | 12.558343 |
| KAZ_245 | 0.61832537 | -50.927949 |
| KAZ_247 | 0.16096904 | -36.014223 |
| KAZ_46 | 0.67971115 | 2.25446858 |
| KAZ_207 | 0.54521529 | -47.141467 |
| KAZ_208 | 0.60108875 | -9.4716777 |
| KAZ_241 | 0.69988563 | -155.79388 |

| | | |
|---|---|---|
| **KAZ_4** | 0.05372463 | -2.1409364 |
| **UZB_41** | 0.71044366 | 11.0340178 |
| **UZB_10** | 0.75050692 | 9.03011076 |
| **KAZ_209** | 0.49817945 | -3.182834 |
| **KAZ_211** | 0.28370087 | -7.0352234 |
| **KAZ_219** | 0.39149531 | -29.735144 |
| **KGZ_1** | 0.1907625 | -5.8348122 |
| **KGZ_2** | 0.38130093 | -1.2687055 |
| **KGZ_4** | -0.3835976 | 12.2549014 |
| **UZB_6** | -0.1466292 | 7.76193374 |
| **UZB_26** | 0.07307469 | 11.1465427 |

**Section 5.1.2**

As before, the introductory paragraphs should be part of the methodology section.

R: Thank you for your suggestion. We will incorporate the introductory paragraphs into the methodology section.

Then, it is not clear how the 2005 event is used in calibration. As I note in the paragraph below, the authors apparently compute observed loss in "current" value incorrectly, therefore spoiling the whole calibration. Then again, not sure how it feeds into calibration.

R: We understand the confusion surrounding the utilization of the 2005 event in our calibration process. To clarify, the 2005 event was solely utilized for the validation of flood extent and total losses, not for calibration. This event was chosen due to the availability of reported losses, flood footprints, and river flow time series, allowing for a direct comparison between modeled and observed losses. However, calibrating the entire model solely based on a single comparison between modeled and observed losses would not constitute a robust calibration strategy.

We acknowledge that the current presentation of this section may have caused ambiguity, and we apologize for any confusion it may have caused. In the revised manuscript, we will ensure to distinctly differentiate between sections regarding validation and those regarding calibration. This will help provide clarity on how each aspect of the model evaluation process was conducted.

The authors write about "trending" and "trended reported losses". I suppose you mean price inflation and deflated reported losses. Then, the calculation is wrong – it apparently applies the typical error of using foreign-currency losses and applying a local deflator or

vice versa. In case of Tajikistan, high inflation is matched by loss of the value of the local currency (somoni) relative to the US dollar. Hence, the 2005 losses will only be 8-11 million USD in 2022 using the consumer price index, or 11-14 million USD using the GDP deflator (using data from IMF's World Economic Outlook). Therefore, the result wouldn't be far from the modelled result. Unless what the authors did is to exposure-adjust the 2005 losses, which would be somewhat consistent with a 4-fold increase in Tajik GDP since then (in US dollar terms).

R: Thank you for your detailed analysis and hypothesis regarding our calculation of "trending" and "trended reported losses." We appreciate your insights into the potential effects of inflation and currency devaluation, particularly in the context of Tajikistan's economic situation. We will thoroughly review our calculation methodology in light of your hypothesis and verify our results accordingly.

The same goes for the subsequent analysis of 7 events (Table 3), hence a much better and clearer description how observed losses were adjusted to be comparable with the model, including original local-currency losses and the adjustments made.

R: We acknowledge the need for a clearer description of how observed losses were adjusted to be comparable with the model. In the revised manuscript, we will provide a more detailed explanation of the adjustments made, including the original local-currency losses and the conversion process.

Finally, authors should reduce the extensive discussion of data issues and leave it for the methodology and discussion sections.

R: Thank you for your feedback. We will condense the discussion of data issues and appropriately distribute it between the methodology and discussion sections.

**Section 5.2**

In contrast to other sections, there is very little comment on the figures, especially in 5.2.2, particularly in contrast to extensive descriptions in 5.3.

R: Thank you for your observation. We will ensure to provide more detailed commentary on the figures in section 5.2.2, aligning with the level of description seen in other sections, especially in comparison to the more extensive descriptions found in section 5.3.

**Section 5.3**

A lot of the information in this section repeats the methods, or should be included in that part of the paper. Otherwise, there are 3 tables here showing the details on different scenarios. They should be rather in the supplement, while a table (or graphs) should contrast the scenarios with each other. Further, the use of per mille should be rather replaced by percentages (also next to numbers), making the results more self-explanatory. Finally, the authors lump together in the 2080 scenario the effect of climate change and exposure change (for one sector only). The authors should present those effects separately, and the exposure scenario preferably with contrast to the 'present-day' losses pertaining only to the residential sector.

R: We presented the future scenario projections with the aggregate effect of climate change and exposure mainly to keep the paper as simple and short as possible, bearing in mind that this is a rather long and complex paper. We would prefer to keep it like this, for this reason and also because we believe all the future changes should be accounted for at the same time, since both climate and exposure changes are happening at the same time. In previous experiences, we have found that presenting impacts of different changes separately can be misleading.

Also, the authors suddenly mention here results of earthquake risk, which is not the topic of the paper. Related text should be moved to the discussion.

R: Apologies, we will eliminate the part discussing earthquake risk.

**Section 6.1**

This part contains information that should be part of introduction (motivation of the study) or conclusions.

R: Thank you for your feedback. We will consider relocating the relevant information to either the introduction or conclusions sections, as appropriate.

**Section 6.2**

This section should be much expanded with elements that are currently in other parts of the paper (as mentioned above in the review). It (and 6.1) shouldn't be in a list format, but as plain text. Information about availability of the data should be in the "Data availability" section in the end.

R: We acknowledge the need to expand this section by incorporating elements currently dispersed throughout the paper, as highlighted in your review. We will revise sections 6.1

and the mentioned parts accordingly, presenting the information in plain text rather than a list format to enhance readability and coherence. Additionally, we will ensure that details regarding data availability are appropriately consolidated into the "Data availability" section at the end of the paper. These adjustments will streamline the presentation of information and improve the overall clarity of the manuscript.

**Geographical names**

In the paper, the authors apply the geographical terminology inconsistently. Spelling mistakes and incorrect names are multiple. The authors should consult, in particular, the ISO 3166-1 and 3166-2 standards. Consequently:

- For consistency, use "Kyrgyzstan" short name from ISO rather than the long name "Kyrgyz Republic", as you use short name for the other countries;

- Use "Region" rather than "Oblast". The latter is a Russian name used only in two out of five countries in the study, so its use is inappropriate. ISO 3166-2 uses "Region" as the English name for the first-order administrative units of all five countries.

- Check the spelling of all regions mentioned in the paper according to ISO 3166-2, which provides the correct forms for the national languages of each country.

- The capital of Kazakhstan is called "Astana" again, since 2022.

- Region codes in figures 17-19 sometimes follow ISO 3166-2 codes, but mostly not. Please correct this according to that standard for consistency and to facilitate reuse.

R: We thank the Reviewer for these comments on the geographical names of the project countries. We fully agree on the importance to be consistent with the ISO standards. However, the geographic names used in this manuscript are consistent with the official documentation provided by the World Bank (e.g. project Terms of Reference, GIS maps with the names of the administrative regions, oblasts and so on), reports delivered as part of the project and results (i.e., deliverables) that will be publicly available via a World Bank portal, as well as all the companion papers published in this Special Issue and related to this SFRARR project (e.g. the exposure papers by Scaini et al, 2023 and 2024; the seismic risk paper by Salgado et al., 2024, the seismic hazard papers by Poggi et al., 2024a and b). All results, presented in tabular format, maps and graphs, refer to the geographical names used in this manuscript. We, therefore, prefer not to update the names used in this document to maintain consistency with the project deliverables and other published results.

**Terminology**

The authors' use of "Large-scale" is problematic. Though the term is by now commonly used in papers, it is imprecise. In geography, it has actually reverse meaning (a large-scale map

covers a very small area). In one place, the authors define it as "hundreds of thousands of square km", though the study area covers 4 million km², in another they call it "country-scale". Given the size of the study area, which is comparable with the entire European Union, the heavy use of global datasets, and the moderate resolution of meteorological inputs (0.1°) and the flood maps (3"), I strongly suggest for the authors to replace "large-scale" with "continental-scale" in the title and through the whole manuscript. That would introduce precision and improve the visibility of the paper.

R: We thank the Reviewer for all the notes and suggestions on how to improve the manuscript contents. In this case, however, the suggestion of using "continental-scale" in the title is misleading from our point of view because we are not talking about a continent but about five countries, very large yes, but which alone do not represent a continent.

The authors use "AAL" but the more common acronym is EAD (expected annual damage).

R: Noted.

**Minor**

L88: "global" rather than "regional" (cf. section 3.1)

R: Noted.

L292: how was the 0.1° climate data made to fit the 1 km grid of the hydrological model? The latter must use some geographical projection, correct?

R: The hydrological model uses an equal-area projected reference system. Rainfall values have been associated to each of the hydrological model grid using a simple nearest neighbour methodology.

L322: "0.00083° " would be somewhat better called "3 arc seconds"

R: Noted.

L378: "time horizon 2080" -> what does that mean, exactly? Normally, climate projections look at a particular window, e.g. 2071-2100.

R: Yes, we used the 2071-2100 time window indeed.

L486: where parameters related to lakes and reservoirs part of the calibration?

R: No, parameters related lakes and reservoirs were not calibrated. In this region, several large reservoirs exist, and have a significant impact on the flow peaks. Setting up reservoir models was not an option due to lack data regarding bathymetry, spillways, outlets and management rules. On the other hand, flow data were typically available for stations downstream large reservoirs. Therefore, for river reaches downstream reservoirs, we opted for adjusting the modelled extreme value distributions based on the observed extreme value distribution, by decreasing the modelled peak flows by a factor equal to the ratio of observed and modelled average annual flow maxima.

Fig. 10: it is pretty pointless making a figure just to display a single data point per panel.

R: Thank you for your feedback. We will remove figure 10 and report the values directly in the text for easier comparison.

Authors should leave statements like the ones in L600-L602 for the conclusions.

R: Thank you for the suggestion, we will move this paragraph accordingly.

Fig. 14: 4 out of 5 are national capitals, but not in case of Turkmenistan. Is Ashgabat not at risk? Also, the axis with the return period should be logarithmic for better presentation.

R: Thank you for your observation. While Ashgabat is indeed the national capital of Turkmenistan, it is not prominently at risk of fluvial flooding as it is not situated near any major river. Instead, it faces more susceptibility to pluvial flooding. Turkmenabat, another significant urban center in Turkmenistan, is located along the Syr Darya River and is thus more prone to flood risk. To better represent the flood risk landscape, we focused on the bigger cities most vulnerable to fluvial flooding in each country.

Fig. 15: as above, logarithmic scale is needed, plus a general improvement of the quality of the graphs.

R: Thank you for your feedback. To also address the other reviewer's comment, we will remove Figures 15 and 16 as they are redundant with their corresponding tables.

Figures in general: some figures provide a scale, some not. It should be included in all of them and in general they should be, preferably, made more homogeneous in appearance. Also, if the authors use some kind of background image from external source, it has to be credited. If it's simply the global DEM from Table 1, it still should be mentioned.

R: Thank you for your feedback. We will ensure that all figures include a scale for consistency and improve the overall homogeneity of their appearance. Additionally, any background images from external sources will be appropriately credited.


**Comments by Reviewer n. 2 and Replies by Authors**

**General comments**

This manuscript presents a probabilistic assessment of fluvial flood risk for the countries in the region of Central Asia. My opinion is that the work is a relevant contribution to improve flood risk knowledge in Central Asia, providing a consistent transboundary risk assessment for all the region.

The overall methodology is appropriate for the task and makes use of well-established models and datasets, integrating where possible global-scale a local-scale data. The inclusion of several risk parameters in the analysis (e.g. damage for different economic sectors) is appreciable

Having said that, I think that the manuscript needs to be improved in some parts. The descriptions of some components of the methodology are rather short or incomplete and should be expanded. Similarly, the presentation of results should provide a more detailed overview of the different outcomes.

R: Thank you for your assessment of our manuscript. We appreciate your acknowledgment of its relevance in improving flood risk knowledge in the region, particularly through the provision of a consistent transboundary risk assessment.

We're glad to hear that you find the overall methodology appropriate.

We take your suggestions for improvement seriously. We recognize the need to expand upon certain components of the methodology that may be lacking in detail and to provide a more comprehensive overview of the results. Your feedback will guide our revisions to ensure the manuscript meets the standards of clarity and completeness expected by readers.

**Specific comments**

**Abstract**: it is not well structured as it is now. Background information (lines 22-36) should be reduced or moved in the introduction (for instance, the work done in the project on pluvial flood and seismic risk). At the same time a short summary of the main elements of the methodology and main results should be added.

R: Thank you for your feedback on the abstract. We will restructure it to reduce background information and potentially move it to the introduction. Additionally, we will include a concise summary of the main elements of the methodology and key results to improve clarity and organization.

**Figure 1**: please add country names and, if possible, other details such as the location of main rivers (in particular those mentioned later in the text) and/or main urban areas

R: Thank you for your suggestion regarding Figure 1. We will incorporate country names and, if possible, include additional details such as the location of main rivers and major urban areas.

**Page 4 L135-137**: can you please explain how you used observed data from the KNMI Climate Explorer to correct ERA5-Land extreme precipitation estimates?

R: Regrettably, the bias correction of ERA5-Land precipitation using raingauge data is not relevant to the present paper and should not have been placed in the draft originally submitted. The original study carried out fluvial and pluvial flood risk assessments, but the present paper only shows fluvial risk assessment. This bias correction mentioned here was not carried out for the fluvial risk assessment, only for the pluvial risk assessment, and therefore is not relevant here. We apologise for the confusion. We will remove all references to pluvial flood and ERA5-Land bias correction from the paper.

**Figure 2** is not relevant within the description, so I suggest removing it

R: Noted.

**Figures 3 and 4**: I think they can be merged

R: We agree with the reviewer and will merge the two figures.

**Page 5**: please provide a reference for the GlobeLand30 dataset. Is it based on static or dynamic-over-time land use information?

R: GlobeLand30, developed by the National Geomatics Center of China, is a global land cover dataset that categorizes the earth's land into 10 classes worldwide, including water bodies, wetlands, artificial surfaces, cultivated land, forests, shrublands, grasslands, bare land, tundra, and permanent snow & ice. This dataset offers a resolution of 30 meters and it provides information for two time points (2000 and 2010) for comparison. The dataset has been independently assessed to be 83% accurate (Dong, et al. 2015). The data is available for non-profit use, and it was contributed to the UN in September 2014 (https://www.un-spider.org/links-and-resources/data-sources/land-cover-map-globeland-30-ngcc).

In the revised manuscript we will include a reference and a brief description of the dataset.

**Page 5 L 144** "This dataset has been made accessible to the United Nations (UN) through the UN-ESCAP Statistics Division and the UN_ESCAP ICT and Disaster Risk and Reduction Division": please move this detail in the Data Availability section.

R: Thanks for the suggestion, we will move it accordingly.

**Page 5 L 150** the reference for the original MERIT DEM should be Yamazaki et al (2017)

R: Noted.

**Page 5 L 163-168**: I think that the details on the method applied to combined local-scale information on flood defences with the FLOPROS, WorldPOP and Landsat HBASE datasets (now in page 11 L333-348) should be moved here.

R: We agree. Thank you.

**Page 9**: the description of the CA2D model is perhaps too long compared to the description of the other models

R: Thank you for your observation. We will review the description of the CA2D model and ensure it is balanced with the descriptions of the other models for consistency.

**Pagge 9 L250-254**: I suggest moving this description in Section 3.2. Also, can you provide more information on the exposure database? For instance, does it include building-scale or aggregated information? Which infrastructures are considered? In the following sections you name different types of infrastructures, they should be described here

R: All the details related to the developed exposure model and the assets at risk included in the model can be found in two companion papers (Scaini et al., 2023 – for population and residential buildings; Scaini et al., 2024 – for non-residential buildings, transportation infrastructure and cropland).

**Page 11 L301**: Generalized Extreme Value

R: Ok. Thank you.

**Section 4.1** is quite long. Consider splitting it into two or more subsections (e.g. hydrological modeling, hydraulic modelling and stochastic analysis)

**Figure 5** is not much useful so it could be deleted.

R: Thank you for your suggestion. In order to address both your and the other reviewer's comments, we will revise Section 4.1 and reorganize it into subsections corresponding to the 5 steps. Since each step will correspond to a specific portion of Figure 5, retaining the figure will aid in streamlining the discussion.

**Page 11**: did you undertake any calibration of the CA2D model (e.g. the roughness parameters)? Also, how did you identify the river sections mentioned in the description?

R: Thank you for your inquiry. The calibration of the CA2D model primarily focused on reproducing historical event hydrographs in terms of volume and peak timing. Specifically, our calibration efforts centered on adjusting configurations to best replicate historical event hydrographs, with the final assumption of a triangular hydrograph reaching the flood peak at 2/3 of the concentration time and returning to zero at twice the concentration time.

We did not calibrate roughness parameters. Instead, we relied on roughness coefficients obtained from literature sources (Arcement, et al., 1989).

**Page 11 L333-348**: see my previous comment about the opportunity of moving this description to Section 3.1. Besides that, how did you implement flood protections in the risk modeling framework? Did you explicitly include flood protections in CA2D simulations (e.g. by modifying the DEM) or only in the risk analysis (i.e. assume that the all floods below the design level would cause no damage)?

R: Thank you for your feedback. We will relocate the description to Section 3.1 as suggested.

In our risk modeling framework, flood protections were not explicitly accounted for in CA2D simulations. Instead, we adjusted water depth maps to reflect the presence of flood protections. Specifically, we assumed that areas with water depths below the designated level of protection would not incur any losses.

**Page 12 L375-383**: there is little information on the climate scenarios. Could you please describe the underlying models that were used to define these scenarios (GCMs. RCMs) and provide references? How did you downscale future climate scenarios to match ERA5-land data resolution?

R: The climate scenarios we employed are described in Ozturk et al. (2017). In that stud, the Regional Climate Model (RCM), RegCM4.3.5 of the International Centre for Theoretical Physics (ICTP) was driven by two different CMIP5 Global Climate Models (GCMs), the HadGEM2-ES GCM of the UK Met Office Hadley Centre and the MPI-ESM-MR GCM of the German Max Planck Institute for Meteorology, under two emission scenarios (RCP4.5 and RCP8.5). Based on its predictive skills, we selected the MPI-ESM-MR GCM. We selected the RCP4.5 over the RCP8.5 as it is more in line with the current emission pathway and the future pledges for emission reduction (Roger Pielke Jr *et al* 2022). The model was run over CORDEX's (Giorgi et al., 2009) Central Asia domain (with the corner points at 54.76°N - 11.05°E, 56.48°N - 139.13°E, 18.34°N - 42.41°E, and 19.39°N - 108.44°E).

Bias-correcting climate projections before using them in hydrological modelling is standard practice and should always be carried out to avoid propagating the climate model biases into the hydrological model results (Shrestha et al., 2017; Teutschbein and Seibert, 2012). The methodology we used here belongs to the "delta change" family cited by Teutschbein and Seibert (2012), The literature on this methodology and its implications on hydrological model outputs is very extensive and well documented, here we cite only a few examples (Räty et al., 2014; Mudbhatkal and Mahesha, 2017; Räty et al., 2018; Fang et al., 2015). It is simpler than other techniques, since it does not require to bias-correct the baseline climatology (which is still the observed climatology), although it has the disadvantage that some properties of the variable to be corrected still remain unadjusted (for example, if the precipitation from a certain climate projection is simply multiplied by a factor in order to reproduce the annual average of the reference dataset, the distribution of the original reference dataset will be maintained and only the mean values will be corrected – this is also called "constant scaling"). However, the approach used in this paper, which adjust the whole distribution of precipitation and temperature, not only the mean or the standard deviation, limits this disadvantage. Räty et al (2014), among others, have discussed the advantage and disadvantages of such technique, which blends the simplicity of the delta factor methodologies with the robustness of the quantile mapping methodologies.

**Section 4.2**: some comments here: 1) there seems to be no description of how you applied SSP scenarios in the risk analysis, please provide details. 2) provide a complete list or table of all the risk parameters (impact for different economic sectors, population affected, mortality etc.). 3) put the description of calibration/validation in a dedicated subsection 4) it is sometimes difficult to understand which data were used for calibration and validation of the model (here and in Section 5 too), please try to make it clear.

R: Thank you for these comments. All the details related to the considered SSPs and their application in the exposure model development and therefore in the risk analysis can be found in the paper Scaini et al., 2023.

**Page 13 L430**: you mention a list of historical events and reported losses, can you provide a reference for this information, is it coming from local governments?

R: The list of historical reported losses is sourced from local governments/agencies and was obtained as part of our project's objective of regional cooperation and capacity building. In the revised manuscript, we will explicitly mention the data source to provide proper credit and transparency.

**Section 5.1.1**: change section name to "hydrological model"

R: Noted.

**Figure 7**: make sure that the Y axis for precipitation and temperature data is readable in all panels. Also, the small maps on the top left side of each graph are difficult to interpret (is it the river basin?). Perhaps it could be useful to put a separate map to locate the different river sections in the region.

R: Thank you for your feedback. In response to other reviewer suggestions, we plan to enhance this section by providing more detailed information on the hydrological model calibrated parameters and incorporating relevant metrics from the stochastic evaluation.

To improve clarity, the figures will be moved to the supplementary material, and we will take measures to enhance their readability as suggested.

**Figure 8**: the trend lines should be more visible

R: Noted.

**Section 5. 1.2**: please put in dedicated sections the calibration of vulnerability curves and the validation of flood extent maps (they should be presented before the validation of modelled risk estimates).

R: Thank you for your suggestion. This section currently lacks clarity and will be reorganized to distinguish between calibration and validation procedures and results.

**Section 5.1.2**: Here you mention vulnerability functions for infrastructures and crops, they should be described in Section 4 (I undestsnf they are taken from previous studies but you should specify, for instance, if you used separated functions for each country).

R: The assets covered are buildings, infrastructure (roads and airport) and crops. More information can be found in a companion paper (Scaini et al., 2023, 2024).

[revised manuscript text omitted]

**Section 5.1.2**: Can you describe how the observed flood extent for the 2005 event was derived? Did you carry out a quantitative comparison of observed and modelled flood extent model?  It would be useful to calculate some performance metrics (see Alfieri et al 2014 for instance) and check if the observed underestimation of impacts might depend on underestimation of flood extent. Also, please include the reported the flood footprint in the map in Figure 9.

R: The 2005 Hamadoni flood is the only event for which reported losses, flood footprints and river flow time series are available.

The flood in Hamadoni, was the subject of the PhD Thesis of one of the local partners, as well as the focus of a report from JICA (Study on Natural Disaster Prevention in Pyanj River,

2007, https://openjicareport.jica.go.jp/pdf/11870748_01.pdf). The observed flood extent as well as the inlet discharge data were extracted from the JICA report.

The flood event was quite long and involved a dyke breach and extensive damage to the nearby villages The JICA study provides both a probable inundation area that was estimated from satellite data, and a flood footprint that was simulated. The following factors contribute to the uncertainty in both the JICA results and ours:

- Satellite data from SPOT and ASTER are only available before and significantly after the peak. This probably explains the satellite estimated flood map underestimation of the flood extent.
- The inlet discharge was estimated from the data recorded at a different station which is located at 80 km upstream, by the peak discharge ratio.
- The simulated water depth values are not available from the JICA study, we only have a figure that we superimposed over our flood footprint for a visual comparison.
- We did not simulate the dyke breach as information on its location and the nature of the damage was not available.

We built our hydrograph by taking the data estimated by JICA and used it as input to the CA2D model to get the flood footprint for the event.

In the revised manuscript we will provide both reference to the original study and the figures of the visual comparison.

**Comparison with the satellite estimated flood map**

[Figure]

**Comparison with the JICA flood map**

[Figure]

**The graphs in Figure 10** are not useful because there's only one point for each graph. I would replace them with a table or directly include the numbers in the text. In case the event is reported in global loss datasets (for instance, in the International Disaster Database EM DAT, https://www.emdat.be/) this could be used as an additional reference for comparison.

R: Agreed, we will provide the numbers in the text and provide a reference.

**Page 19 L595**: do you mean that you have calibrated the loss model based on this comparison? Or are you referring to the calibration done on the vulnerability functions mentioned before?

R: Thank you for bringing this to our attention. We apologize for any confusion caused by the lack of clarity in this section regarding calibration and validation. Indeed, the reported losses were utilized for both the validation of the entire modeling chain and the calibration of the vulnerability functions.

In the revised version of the manuscript, we will reorganize this section to clearly distinguish between calibration and validation processes for the various model components. This will ensure better clarity and accuracy in describing our methodology.

**Page 20** Possible explanations for underestimated impacts are that pluvial flooding impacts were not considered, as well as impacts ng in the minor drainage network not included in hydraulic modelling.

R: Thank you for raising these points. It is indeed possible that the underestimated impacts could be attributed to several factors, including the exclusion of pluvial flooding impacts and impacts within the minor drainage network not covered in the hydraulic modeling. Additionally, the consideration of landslides could further contribute to the estimated impacts. Furthermore, it's important to acknowledge the significant uncertainty inherent in the process of actualizing damages to current monetary values.

**Figure 14**: the graphs in this figure are not much informative because water depth changes in each pixel of the model; instead, showing flood extent graphs over an area (a country or a river basin) would be more useful.

R: Thank you for your feedback. Figure 14 should provide a comparison in risk profile for 5 representative sites (one for each country), while Figure 12 displays flood extent and intensity over Uzbekistan. We agree that Figure 12 has very small details, in the revised manuscript we could provide a different example of fluvial flood hazard map that includes a smaller area.

**Section 5.3.2**: The Global Assessment Report on Disaster Risk Reduction 2015 (GAR2015) produced risk profiles for all countries in the world for different natural hazards including floods. I think it would be interesting for the reader to see how GAR estimates compare with those of the manuscript (country profiles are available at https://www.preventionweb.net/english/hyogo/gar/2015/en/home/data.html ).

R: Thank you for your suggestion. While comparing our flood risk estimates with those from the Global Assessment Report on Disaster Risk Reduction 2015 (GAR2015) could offer valuable insights for readers, it falls outside the scope of the paper. Nonetheless, we appreciate the idea and will keep it in mind for future research or supplementary analyses.

**Section 5.3.2**: Table 4 and figure 15 show more or less the same information so I would keep the figure in the text and perhaps move Table 4 in a supplement. The same applies for Table 5 and figure 16.

R: Thank you for the feedback, we will only keep the tables in the revised manuscript.

**Table 6**: Comparing Tables 4, 5, 6 it seems that the absolute losses decrease in the 2080-SSP1 scenario, whereas relative losses increase. Can you explain this behavior? Did you observe a similar behaviour in other future scenarios? You might also want to check if your results are consistent or disagree with existing global studies (e.g. Dottori et al 2018, with apologies for the self-citation).

R: The increase in the absolute values of losses is mainly driven by an increase in exposed value. It is widely accepted that increases in exposed value are the main driver of increased losses from natural disasters globally (Pielke, 2021), and this region of the World shows a similar behaviour. The decrease in the relative value of losses is driven by a decrease in flood hazard, caused by climate change. This is mainly due to a reduction in snow cover during winter and, subsequently, a reduction in snow melt in spring and summer, which is one of the drivers of floods in this region. It should be noted that the patterns of climate change-driven effects on flood hazard are highly variable in space, with even some increase in flood hazard in the driest parts of the area, where an increase in precipitation intensity should increase flood hazard. It should also be noted that the uncertainty in such forecasts is very large, and, for a proper assessment of the impacts of climate change at such a distant horizon, other scenarios/models should be used. This was not done within the framework of this study, as it was not the main aim of the project, but it is definitely worthwhile as a follow-up activity.

**Section 5.3.2**. Currently this section only shows results for overall economic damage. I would recommend to provide an overview of all the results produced by the analysis (e.g. human fatalities, breakdown of damage per each economic sector considered, impact on infrastructures etc.). This information would be useful because it is rarely reported in similar studies in literature. Also, you should include in the discussion the results for future scenarios other than 2080-SSP1 (if deemed important, additional results could be added in a Supplement).

R: Thank you for this suggestion. We will introduce more results in the revised version of the manuscript.

**Data availability**: Please provide here the details for accessing all the datasets used in the study (or explain why they are restricted). For instance, several global datasets are freely available (Table 1),

R: All the datasets belong to the World Bank group. However, the Bank has in plan to share the collected data as well as the computed results via a website where all the information can be download by any user.

**Section 7**: you could include a short summary of the main outcomes (e.g. countries with higher relative impacts, risk hotsposts).

R: The revised manuscript will include a short summary of the main outcomes in the conclusion section.

---

## Author Response (AR2)

**Comments by Reviewer n. 1 and Replies by Authors**

**General comments**

The authors have largely properly addressed comments of the review, however there are some exceptions, listed below:

REPLY: Thank you for your review. We have replied to the comments here below.

Section 5.4.1: despite writing in the response that the authors will add information on adjustment of losses, it was not provided.

REPLY: Apologies for that, we have expanded the description on adjustment of losses.

Section 5.4.2: the authors clearly don't understand the issue and seem unable to interpret economic data, and the revision only adds confusion. If the authors only consider change in price levels, then the information that should be used inflation (based on GDP deflator) in Tajikistan between 2005 and 2022 was 410% (factor of 5.1), while the local currency lost 71.8% of its value compared to the US dollar (factor 0.282). Multiplication of those two factors gives the correct adjustment factor (1.44), i.e. 44% increase. If the authors adjust from both price change and economic growth, then the correct approach is to adjust the 2005 value by change in nominal GDP expressed in US dollars, which increased from 2.311 bln USD in 2005 to 10.493 bln USD in 2022, i.e. a factor of 4.54. The information the authors in paragraph supposedly based on IMF data are incorrect. Please carefully recheck your economic analysis. Also, results section uses 2020 exposure, so why 2022 here?

REPLY: While we acknowledge that there might have been some misunderstanding between the reviewer and the authors regarding this topic, the sentence "the authors clearly don't understand the issue and seem unable to interpret economic data" seems out of place in a paper review, especially when it is directed to 16 respectable professionals of the risk assessments, and given that the authors have manifested a strong willingness to implement all the changes and suggestions of the reviewer in the first round of review.

We have modified the section using the data provided by the WB GDP deflator for GDP and by xe.com for currency rates against USD, and using values for 2020 instead of 2022. Apologies for the typo (i.e., "2022" instead of "2020"). We have removed the reference to IMF data.

Section 6.3: it's fine that authors want to keep their combined scenario. But there are other comments on the section that were not implemented.

REPLY: Our apologies. Here your comments and our replies:

*A lot of the information in this section repeats the methods, or should be included in that part of the paper.*

We believe we have addressed this already in the previous round of review.

*There are 3 tables here showing the details on different scenarios. They should be rather in the supplement, while a table (or graphs) should contrast the scenarios with each other.*

We have removed two of the three tables and added a figure instead.

*Further, the use of per mille should be rather replaced by percentages (also next to numbers), making the results more self-explanatory.*

Done.

Section 7.1 and 7.2: no changes were made based on my review

REPLY: Regarding the "Strengths" section to be placed in the introduction or conclusion, respectfully, we do not agree with this comment, and we decided to keep this part in the discussion. We think that identifying strengths and limitations of a study is a rather standard part of a paper discussion section.

Regarding expanding the "Limitations" section (also) with parts from other sections of the paper, this has already been done in the previous round of review. Same with respect to "Information about availability of the data should be in the Data availability section in the end.". We have removed a link to the World Bank repository.

Lastly, we do not see the benefits of putting the bullet-point list of strengths and limitations in plain text. Rather, we think it is clearer as a bullet-point list.

Geographical names: the naming convention used by authors elsewhere or by the World Bank is not an argument. The names should conform with the standards and conventions of field and journal. Please apply the changes as in my review.

REPLY: We have replaced Oblast with region, also following the suggestion of the other reviewer.

We have not found any NHESS journal guidelines on geographical naming conventions other than a generic reference to "United Nations naming conventions", which to us seems consistent with the naming in our paper. If indeed the journal has specific geographical naming conventions which we have missed, please do show them to us. In general, we think that the geographical names we have used are quite uncontroversial and widely used. Can you please provide an example of a name that we have used wrongly and would lead to confusion/controversy?

Aside, we do think that consistency with other companion papers is very important, as well as consistency with the rest of the project guided by the World Bank, whose ultimate goal is to be usable by decision makers. Inconsistency in names among these sources might lead to more confusion than using some other standards.

Furthermore, we have shared the final report of the project (from which this and other papers are stemming) with numerous stakeholders (including government counterparties) from the five countries object of study, none of which has objected to the geographical naming we used. We understand that this is no argument to use in a rebuttal of a review, but it strengthens our belief that the naming conventions used are correct and perfectly understandable.

We also ask the editor to assist us in this matter.

Terminology: here, also please apply the changes as in my review.

REPLY: We have already replied to this comment in the first round of review.

Figures: the authors have decreased, rather than increased consistency of the figures. Please check them according to my final comment of the review.

REPLY: We have included a scale in the maps that were missing one and cited the background images.

**Comments by Reviewer n. 2 and Replies by Authors**

The Authors made a substantial effort to clarify the description of methods and results and to add the details required by the reviewers, so the manuscript reads much better now.

In my opinion, the manuscript still requires some minor edits to improve the clarity of some sections, which I list below. After these have been addressed, I believe that the manuscript can be published.

REPLY: Thank you for your review and for your positive words. We have replied to the comments here below. As some of the comments are simply suggestions for rearranging paragraphs or sections, we have taken the liberty not to implement a few of them, explaining why we did not, mainly for the sake of the readability of the paper. Obviously, there is some subjectivity in how a paper should be structured, so if the reviewer or the editor feel that our decision is not the correct one, please let us know and we will modify the paper accordingly.

Abstract: "The largest relative expected annual damages are found in Kazakhstan and Tajikistan..." relative to GDP? please specify.

REPLY: Relative to the total exposure. We have now specified.

line 40: I would replace "oblast" with "region" here. As a general comment, I saw that the use of Oblast was questioned by one of the other reviewers. If you want to use this term, please provide a clear definition early in the text so that readers can understand it.

REPLY: We have replaced Oblast with region.

lines 42-45: this paragraph would be better placed immediately after lines 25-31.

REPLY: Moved.

lines 320-382: I don't fully get this explanation, could perhaps provide an example for a city?

REPLY: We have simplified the explanation to make it more straight-forward.

lines 612-613: My understanding is that authors performed a sort of manual calibration , but It is not clear which indices were used to assess the model skill , did you use the ones in Table 4, or others? Please specify.

REPLY: We have specified it.

Section 4.1.2 : Lines 631-637 read like a summary of the remainder text in this section, please check to avoid repetitions

REPLY: We have removed repetitions.

Section 4.1.4 I suggest moving the results of validation of modelled hazard maps (i.e. figure 4 and lines 777-779) in section 6.2

REPLY: moved.

Figure 4: Please add a legend on left-hand map. Also, the legend on right-hand map is not readable, perhaps a white background would help.

REPLY: Unfortunately, we do not have an editable version of the right side figure and we can't change the background. We have added an explanation of the colours in the caption.

Lines 874-880: This list is rather long, not much clear and perhaps not necessary. Suggest either to rearrange it or remove it

REPLY: We have removed it.

Lines 924-935: this description would be more appropriate in Section 5.2

REPLY: moved.

Section 5.4 I suggest moving the validation of risk estimates to Section 6.3, as they are part of the results

REPLY: We understand this comment, but in order to avoid a very short risk model validation part in section 6, which would alter the flow of the paper and result a little bit odd, we have finally preferred to keep this part in section 5.4.

Section 6.1 . A map with the correlation and bias values would be more informative. Also, can the authors elaborate on the resulting skill of the model, considering the quasi-continental scale of the model and the consequent limitations?

REPLY: Despite trying, we have not been able to create a map with the necessary quality and informative value to replace Table 4 and to be included in section 6.1. However, we believe that all the information is already in section 6.1. We have added some considerations on the skills of the model.

Lines 1221-1222: this should go in Data Availability section

REPLY: These lines refer to model results, not input data, therefore we think they should be in the results section.

Figure 9: in my opinion the graphs would be more informative with a logarithmic x-axis

REPLY: Here below you can see our best version of the log-log plots. After internal discussion, we think that the non-log version in the paper is better than this one. If the reviewer or the editor think otherwise, we are totally willing to change them.

[Figure]

Lines 1685-1707: this interesting paragraph reads like a discussion of the results of risk model validation (now section 5.4), so perhaps they could be put together.

REPLY: We see this section belonging more to the discussion rather than to the results, so we would like to keep it here.

---

## Author Response (AR3)

**Comments by Reviewer n. 1 and Replies by Authors**

The paper will go through copy-editing, but last remarks: 'oblast' is still left in L622 (tracked-changes version) and Table 3. Also, the unit of the second number is incorrect (should be billions). Also, indeed Figure 9 would work better with logarithmic scale. Please don't forget to rename the 'NurSultan' panel to 'Astana'.

REPLY: Thank you for your review. We have replied to the comments here below.

We have replaced Oblast with region in L516 (previously L622) and in the caption of Table 3.

Regarding the numbers in L538, the unit of the second number is now correct.

Figure 9 has been updated as per Reviewer's request. Furthermore, the "Nur Sultan" panel has been renamed to "Astana".

Regarding the editorial comments, the manuscript has been revised accordingly.

Thank you.